# Bub1 positions Mad1 close to KNL1 MELT repeats to promote checkpoint signalling

Gang Zhang[1], Thomas Kruse[1], Blanca López-Méndez[1], Kathrine Beck Sylvestersen[1], Dimitriya H. Garvanska[1], Simone Schopper[1], Michael Lund Nielsen[1] & Jakob Nilsson[1]

Proper segregation of chromosomes depends on a functional spindle assembly checkpoint (SAC) and requires kinetochore localization of the Bub1 and Mad1/Mad2 checkpoint proteins. Several aspects of Mad1/Mad2 kinetochore recruitment in human cells are unclear and in particular the underlying direct interactions. Here we show that conserved domain 1 (CD1) in human Bub1 binds directly to Mad1 and a phosphorylation site exists in CD1 that stimulates Mad1 binding and SAC signalling. Importantly, fusion of minimal kinetochore-targeting Bub1 fragments to Mad1 bypasses the need for CD1, revealing that the main function of Bub1 is to position Mad1 close to KNL1 MELT repeats. Furthermore, we identify residues in Mad1 that are critical for Mad1 functionality, but not Bub1 binding, arguing for a direct role of Mad1 in the checkpoint. This work dissects functionally relevant molecular interactions required for spindle assembly checkpoint signalling at kinetochores in human cells.

[1] The Novo Nordisk Foundation Center for Protein Research, Faculty of Health and Medical Sciences, University of Copenhagen, Blegdamsvej 3B, 2200 Copenhagen, Denmark. Correspondence and requests for materials should be addressed to G.Z. (email: gang.zhang@cpr.ku.dk) or to J.N. (email: jakob.nilsson@cpr.ku.dk).

A prerequisite for life is the equal distribution of genetic information to the new daughter cells and this requires that the genome is accurately duplicated and subsequently distributed to new cells. Accurate segregation of sister chromatids during cell division depends on the spindle assembly checkpoint (SAC), which in response to improper attachments between kinetochores and microtubules generates a diffusible 'wait anaphase' inhibitor[1–3]. This inhibitor is the mitotic checkpoint complex (MCC) composed of the Mad2 and BubR1–Bub3 checkpoint proteins bound to Cdc20, the mitotic co-activator of the anaphase-promoting complex (APC/C)[4]. The MCC potently inhibits the APC/C-Cdc20 complex and this prevents the entry into anaphase, hereby providing time for proper bi-orientation of all sister chromatids. Once all kinetochores are attached to microtubules, the production of the MCC ceases and, in addition, the existing MCC is disassembled resulting in active APC/C-Cdc20 and the progression into anaphase.

The essential components of the SAC were identified in the early 1990s and include the mitotic arrest deficient (Mad) proteins Mad1, Mad2 and Mad3 (BubR1 in humans) as well as the budding uninhibited by benzimidazole proteins (Bub) Bub1 and Bub3 (refs 5,6). Bub1 and BubR1 are in a stable complex with Bub3, while Mad2 exists in a stable complex with Mad1 and as free Mad2 (refs 7,8). In addition to these structural components, checkpoint signalling also depends on at least three kinases namely cyclin-dependent kinase 1 (Cdk1), Aurora B and monopolar spindle 1 (Mps1)[9–16]. However, only a few phosphosites on kinetochore and checkpoint proteins that directly regulate the SAC have been identified.

A major question in the field is how the checkpoint proteins are recruited to kinetochores and how this stimulates the generation of the MCC. It is clear that all checkpoint proteins localize dynamically to unattached kinetochores, as does Cdc20. This localization depends on a large outer kinetochore complex composed of the KNL1-Zwint complex, the Mis12 complex and the Ndc80 complex (collectively referred to as the KMN network)[17–21]. The KNL1 protein is a direct receptor for the Bub1–Bub3 and BubR1–Bub3 complexes because the phosphorylation of so-called Met–Glu–Leu–Thr (MELT) repeats in KNL1 by the Mps1 kinase generates binding sites for Bub3 (refs 22–30). Bub1 and BubR1 both contain short linear motifs referred to as ABBA motifs (also known as Phe-box or A-box) that act to localize Cdc20 to kinetochores with the removal of the ABBA motif in Bub1 having a more pronounced effect on SAC signalling[31–36].

The exact mechanism behind recruitment of the Mad1/Mad2 complex to kinetochores in humans has not been established, but given the central role of this complex for SAC signalling this is crucial to understand. This contrasts the situation in budding yeast and worms where a direct interaction between Mad1 and Bub1 has been shown to localize Mad1/Mad2 to kinetochores[37–39]. In budding yeast, this Mad1–Bub1 interaction is stimulated by Mps1 phosphorylation of multiple sites in a central unstructured region in Bub1 spanning residues 369–608 (ref. 37). Interestingly, in worms it is the kinase domain of Bub1 that binds Mad1 directly suggesting that the Mad1/Mad2 complex can be recruited to kinetochores by different mechanisms[38]. In human cells a direct interaction between Mad1 and Bub1 stimulated by Mps1 was recently described[40,41] and consistently Bub1 has been proposed to scaffold the assembly of SAC complexes on MELT repeats[30]. In addition the Rod-Zwilch-ZW10 (RZZ) complex is required for Mad1/Mad2 kinetochore localization and checkpoint signalling in higher eukaryotes[33,42–44]. The exact contribution of Bub1 and the RZZ complex is still to be fully dissected.

To further our understanding of Mad1/Mad2 kinetochore recruitment and the function of Bub1 in human cells, we here focus on the molecular function of conserved domain (CD1) in Bub1 that is essential for SAC signalling in both humans and yeast[45,46]. We show here that CD1 is required for a phosphoregulated direct interaction between human Bub1 and Mad1 and that disturbance of this interaction is detrimental to the SAC. In line with this we can bypass the requirement for CD1 by fusing minimal kinetochore targeting Bub1 fragments to Mad1, suggesting that the main function of Bub1 in SAC signalling is to localize Mad1 correctly at kinetochores. Interestingly, residues in a C-terminal globular domain of Mad1 are still required for SAC signalling when Bub1 and Mad1 are fused arguing that Mad1 has additional functions in the SAC. In conclusion, we establish a direct interaction between human Bub1 and Mad1, identify a critical phosphorylation site required for this interaction and identify the key function of the interaction. We hereby provide novel insight into Mad1/Mad2 kinetochore localization in human cells by Bub1 and propose that precise positioning of Mad1/Mad2 close to KNL1 MELT repeats is critical for checkpoint signalling.

## Results

**Bub1 conserved domain 1 is essential for SAC signalling**. The Bub1 checkpoint protein contains different motifs that have been implicated in its function including CD1 and the ABBA motif (Fig. 1a). The function of CD1 is unclear, while the ABBA motif is required for recruiting a fraction of Cdc20 to kinetochores (we will discuss recent parallel data on CD1 (ref. 40) from the Yu lab published during the reviewing of this manuscript in the discussion; Supplementary Fig. 1 and refs 31,32). The removal of CD1 did not affect Cdc20 kinetochore localization suggesting that CD1 performs a different function in the checkpoint (Supplementary Fig. 1). To directly compare the importance of these two motifs, we first established a Bub1 RNAi rescue system in HeLa cells (Fig. 1b). Compared to control treated cells that had a median mitotic duration of 555 min in the presence of low dose of nocodazole, Bub1 RNAi-treated cells only spent 285 min in mitosis, revealing a requirement for Bub1 for efficient checkpoint signalling (Fig. 1c,d). By co-transfecting Bub1-depleted cells with a RNAi-resistant plasmid encoding Bub1 residues 1–553 tagged with Venus, which we have previously shown to fully support SAC signalling in MEFs devoid of endogenous Bub1 (ref. 33), we could restore mitotic duration to the same level as in control cells. Bub1 1–553 localized efficiently to kinetochores as did all constructs in this study unless otherwise specified. Only cells expressing similar fluorescent intensities were analysed and quantified (Fig. 1d; Supplementary Fig. 1). To confirm the SAC deficiency achieved by live cell imaging, we analysed the ability of Mad2 to interact with Cdc20, BubR1 and the APC/C under these conditions by immunoprecipitation of Mad2 and found that Bub1 RNAi resulted in reduced interaction and that this could be fully rescued by expression of Bub1 1–553 (Fig. 1e,f; Supplementary Fig. 11), revealing a role of Bub1 in efficient MCC formation.

Next, we used our Bub1 RNAi complementation system to directly compare the importance of CD1 and the ABBA motif. We transfected Bub1-depleted cells with Bub1 1–553ΔCD1 lacking the entire CD1 (residues 458–476) or with Bub1 1–529 that removes critical residues from the ABBA motif (Fig. 1a). Cells complemented with Bub1 1–529 had a mitotic duration of 370 min, while cells complemented with Bub1 1–553ΔCD1 only arrested for 110 min (Fig. 1c,d). A similar short mitotic timing was observed in Bub1 1–529ΔCD1 expressing cells indicating no additional effect of removing the ABBA motif when CD1 was deleted.

A likely explanation for the inhibitory effect of Bub1 proteins lacking CD1 is that they do not support any SAC signalling and

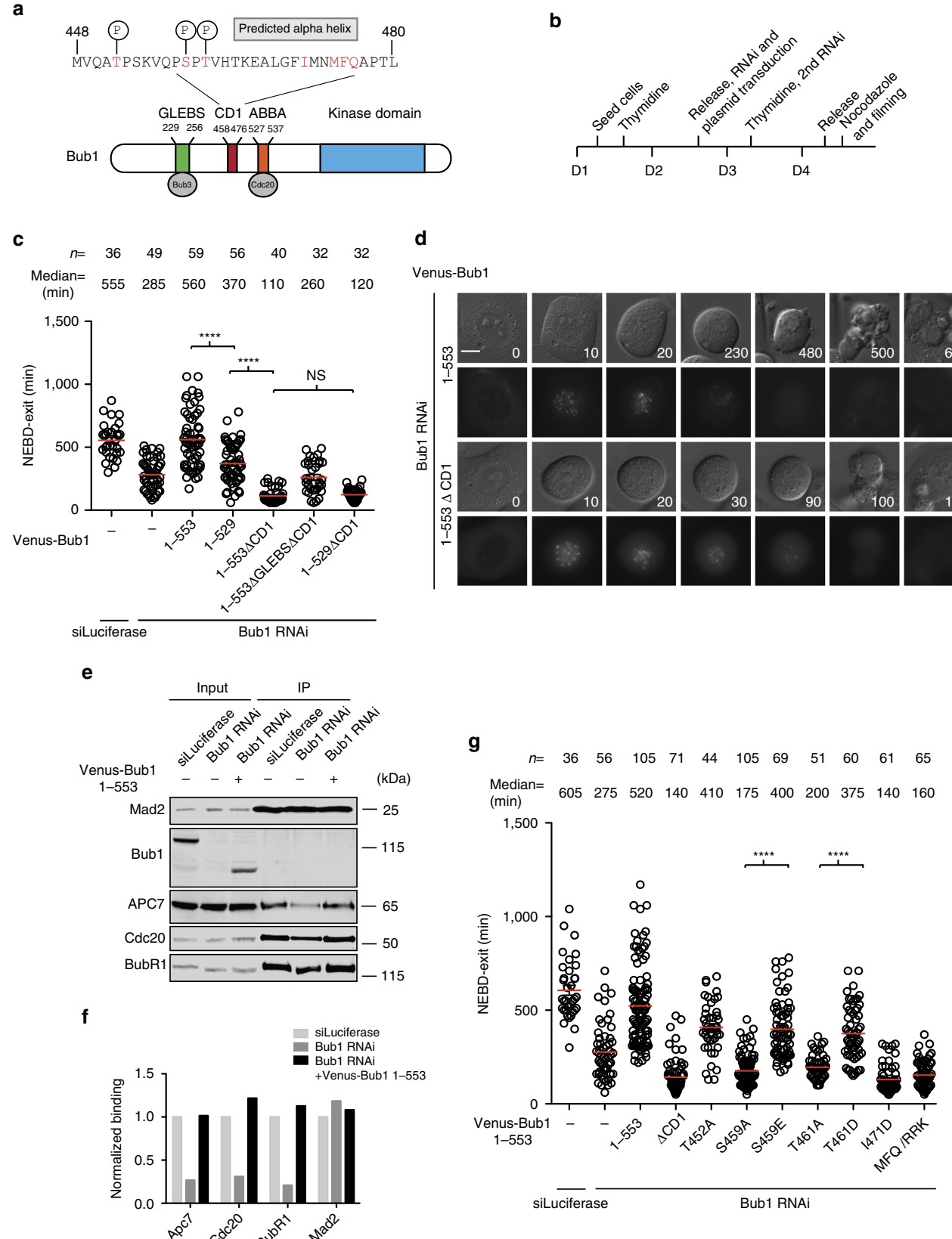

compete with remaining endogenous Bub1 for binding to KNL1 similarly to what has previously been observed with another Bub1 mutant (Bub1 1–311)[47,48]. Indeed, a Bub1 mutant lacking both the CD1 and with mutations in the Bub3 binding domain (referred to as ΔGLEBS) did not localize to kinetochores and showed a very similar mitotic timing as Bub1 RNAi treated cells (Fig. 1c; Supplementary Fig. 2). The mitotic timing of 110 min observed in Bub1 1–553ΔCD1 expressing cells thus likely

represents the timing in cells where endogenous Bub1 is fully inactivated by a combination of RNAi depletion and competition for KNL binding from non-functional Bub1.

The conclusion from these experiments is that CD1 is essential for Bub1 functionality and its removal results in inactive forms of Bub1. Therefore, we set out to understand the function of CD1.

**Phosphorylation sites in CD1 are required for the SAC**. We first analysed the importance of different residues in CD1 for a functional checkpoint. The CD1 contains two reported phosphorylation sites, Ser459 and Thr461 (refs 49,50). Following the phosphorylation sites is a region that is predicted to form an alpha helix[51] and indeed circular dichroism spectra of CD1 peptides confirmed the possibility of an alpha helix (Fig. 1a; Supplementary Fig. 3).

We mutated either phosphorylation sites or hydrophobic residues in CD1 and investigated whether this resulted in inactive forms of Bub1 1–553 arguing that they are critical for Bub1 function. All Bub1 CD1 mutants tested localized to kinetochores similar to wild-type Bub1 arguing that they are correctly folded and bind Bub3.

Mutation of Ser459 or Thr461 to Ala inactivated Bub1 showing that these two residues are critical for CD1 functionality (Fig. 1g; Supplementary Fig. 4A). For comparison, mutation of another reported phosphorylation site just outside of the CD1 region, Thr452, had only a minor effect on SAC signalling. Cells complemented with Bub1 having phosphomimicking mutations, Ser459 to Glu or Thr461 to Asp restored checkpoint activity arguing that phosphorylation of Ser459 and Thr461 in Bub1 is important for SAC signalling. Mutation of hydrophobic residues Ile471 or Met–Phe–Gln (residues 474–476) to charged residues resulted in Bub1 constructs that were as defective as deletion of the entire CD1.

To investigate whether the phosphorylation of Bub1 occurs at kinetochores *in vivo*, we generated a phosphospecific antibody against a peptide phosphorylated on both Ser459 and Thr461 (referred as pSpT in Fig. 2). The antibody stained kinetochores of nocodazole arrested cells, but when cells were depleted of Bub1 or treated with lambda phosphatase before staining the kinetochore staining disappeared (Fig. 2). Furthermore, expression of Bub1 1–553 in Bub1-depleted cells restored the kinetochore staining, but this was absent when Ser459 was mutated and largely reduced when Thr461 was mutated further confirming the specificity of the antibody (Fig. 2d,e). So far, we do not know the relative contribution of the Ser459 and Thr461 phosphorylation sites to the specificity of the antibody. In undisturbed mitotic cells, the antibody stained kinetochores of cells in prometaphase, but reduced kinetochore staining was observed when cells were in metaphase and no staining was observed on anaphase kinetochores (Fig. 2c).

To identify mitotic kinases able to phosphorylate Bub1 on Ser459 and Thr461 we performed *in vitro* phosphorylation assays with candidate kinases. The Ser459 phosphorylation site matched a Cdk1 consensus, while the Thr461 did not resemble any known kinase consensus. However, given the fact that Mps1 phosphorylates the corresponding Thr455 site in budding yeast Bub1 (ref. 37) we suspected that Mps1 might be the kinase that phosphorylates Thr461. We incubated purified GST-Bub1 encompassing amino acids 425–500 with either purified Cdk1-Cyclin B1 or Mps1 and analysed the samples by mass spectrometry. Addition of Cdk1-Cyclin B1 resulted in phosphorylation of Ser459 and Thr461, while addition of Mps1 resulted in phosphorylation of Thr461 but no phosphorylation on Ser459, supporting that these kinases are able to phosphorylate functional relevant phosphorylation sites in CD1 (Supplementary Fig. 4B).

In conclusion, phosphorylation of functional relevant sites in CD1 occurs at kinetochores *in vivo* and the *in vitro* phosphorylation assays indicate Cdk1–Cyclin B1 and Mps1 as possible mitotic kinases able to phosphorylate CD1. In addition, hydrophobic residues in CD1 are also critical for functionality of Bub1.

**Bub1 conserved domain 1 is required for direct Mad1 binding**. We speculated that CD1 might engage in a protein-protein interaction that is critical for checkpoint signalling. Mad1 appeared to be the only core component of the checkpoint for which we could see an effect of removing the CD1 on its kinetochore levels (Supplementary Fig. 1 for Cdc20 and ref. 33 for BubR1, Mad1). However, an interaction between human Mad1 and Bub1 cannot be detected in immunoprecipitation experiments[52]. To determine whether there was an interaction between human Mad1 and Bub1, we first developed an assay for looking at transient and/or weak Mad1 interactions *in vivo*. To this end, we used a proximity-dependent ligation assay (BioID[53]) based on a fusion of the promiscuous biotin ligase BirA to either the N terminus or the C terminus of Mad1. In stable cell lines expressing BirA–Mad1 or Mad1–BirA at close to endogenous levels the kinetochore was biotinylated upon addition of biotin to the media, and we purified the biotinylated proteins using streptavidin beads under stringent conditions (Fig. 3a–c; Supplementary Fig. 5A,B). The purification and detection of biotinylated proteins were highly robust and we detected multiple Mad1 peptides upon addition of biotin, as expected (Supplementary Fig. 5C). Using a quantitative mass spectrometry SILAC approach allowed us to compare the proximity of proteins to the N terminus versus the C terminus of Mad1. The SILAC ratio of BirA-Mad1 to Mad1-BirA was one with this approach. This showed that Bub1, BubR1 and Cdc20 were more efficiently labelled with biotin in the Mad1-BirA

**Figure 1 | The CD1 domain is essential for Bub1 function.** (**a**) Schematic of Bub1 primary structure with the position of GLEBS, CD1, ABBA motif and kinase domain indicated. Numbers on top refer to residue numbers and amino acid sequence of CD1 and surrounding residues is above the scheme. The residues mutated in this study are indicated in red. Reported human Bub1 phosphorylation sites are indicated above with a P and the region of CD1 forming a potential alpha helix is indicated. (**b**) Outline of synchronization and RNAi depletion protocol used in this study. (D: day). (**c**) Bub1 RNAi and rescue with the indicated Bub1 RNAi-resistant constructs was performed in HeLa cells using the protocol outlined in **b**). A low dose of nocodazole was used for the live cell imaging and the time from nuclear envelope breakdown (NEBD) to exit was measured in single cells and indicated by the circles. The number of cells analysed per condition and the median mitotic timing (red line) are indicated above each condition. A Mann–Whitney test was used for statistical comparison of the different samples. (****$P \leq 0.0001$, NS: non significant). (**d**) Representative still images for Venus-Bub1 1–553 and Venus-Bub1 1–553 ΔCD1. Scale bar, 5 μm. (**e**) Parental HeLa cells or stable HeLa cells expressing inducible Venus-Bub1 1–553 were treated twice with RNAi against luciferase as control or Bub1 before treatment with nocodazole (200 ng ml$^{-1}$). Mitotic cells were collected and Mad2 immunoprecipitated using a pan-Mad2 antibody and the level of associated Cdc20, BubR1 and APC7 was determined by quantitative western blot. Representative of two independent experiments. (**f**) Quantification of experiment shown in **e**. (**g**) HeLa cells were treated similarly as **c**. Live cell imaging was performed in the presence of low dose of nocodazole and the time from NEBD to exit was measured in single cells. Results are presented as **c**. A Mann–Whitney test was used for statistical comparison of the different samples. (****$P \leq 0.0001$).

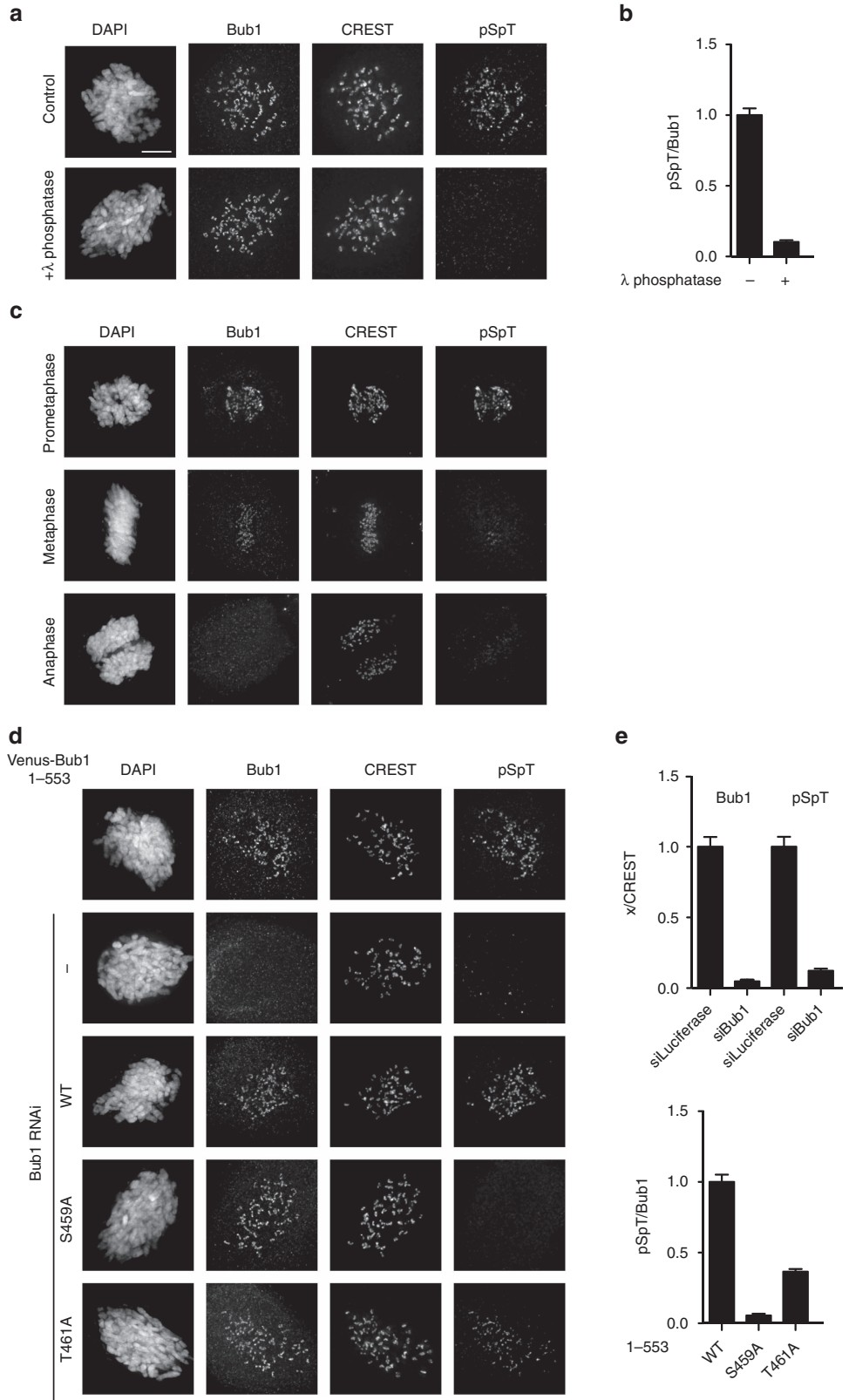

**Figure 2 | Bub1 CD1 is phosphorylated at kinetochores _in vivo_.** (**a**) Nocodazole treated HeLa cells were fixed and treated with $\lambda$ phosphatase at 37 °C for 30 min before staining for Bub1, CREST and pSpT. Scale bar, 5 µm. (**b**) Quantification of kinetochore staining in **a** by pSpT was normalized to kinetochore staining of Bub1. (**c**) Unperturbed mitotic cells were fixed and stained with Bub1, CREST and pSpT antibodies. (**d**) HeLa cells transfected with Bub1 RNAi oligos and RNAi resistant Venus-Bub1 1–553 constructs for 48 h before treatment with nocodazole for 2 h (200 ng ml$^{-1}$). Cells were fixed and stained with Bub1, CREST and pSpT antibodies. (**e**) Quantification of kinetochore staining of Bub1 and pSpT in **d** was normalized by CREST signal (top panel). Quantification of kinetochore staining of pSpT was normalized to Bub1 (bottom panel). At least 200 individual kinetochores from ten cells were measured in each condition from at least two independent experiments. The mean with s.e.m. is indicated. Scale bar, 5 µm.

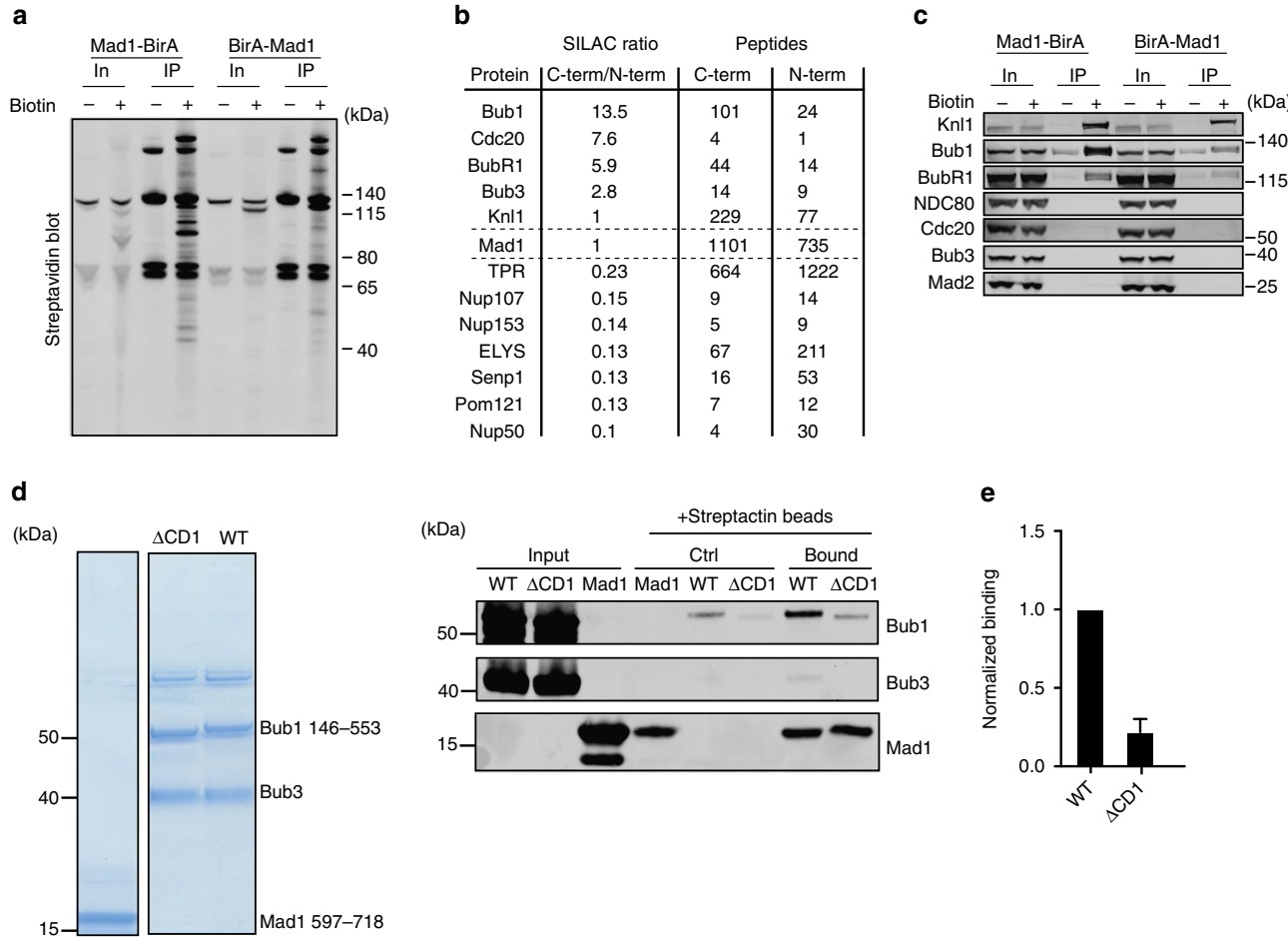

**Figure 3 | Bub1 directly interacts with Mad1.** (**a**) Nocodazole arrested HeLa cells stably expressing the indicated Mad1-BirA or BirA-Mad1 fusion protein at near endogenous levels were analyzed 16 h after induction with doxycyclin with or without exposure to 25 μM biotin. Biotinylated proteins were precipitated with streptavidin beads and visualized by western blot using fluorescently labelled streptavidin. (In: input and IP: purified biotinylated proteins) (**b**) A single SILAC mass spectrometry experiment was performed to determine the ratios of proteins isolated with streptavidin beads from Mad1-BirA versus BirA-Mad1 expressing cell lines respectively. The obtained SILAC ratio for a number of SAC and kinetochore proteins is indicated as well as a number of NPC proteins (**c**) Experiment as in **a,b**, but analysed by western blot for the indicated proteins (representative of at least two independent experiments) (**d**) Strep-Mad1 incubated with FLAG-Bub1 146–553/Bub3 complex (WT) or FLAG-Bub1 146–553ΔCD1/Bub3 complex (ΔCD1) was captured by Strep-Tactin beads and following washing bound proteins were released. Bound Bub1 protein was detected by blotting for FLAG. (**e**) Quantification of three independent experiments as in **d** with WT binding set to 1. Mean and s.e.m. shown.

expressing cells, suggesting that they were closer to the C-terminus of Mad1 (Fig. 3a–c). In contrast, a number of inner nuclear pore proteins were more efficiently biotinylated in the BirA-Mad1 expressing cell line in agreement with the reported interaction of the Mad1 N-terminus and the nucleoporin TPR[54]. Despite Mad2 being a stable binding partner of Mad1 we did not detect labeling of Mad2, which can be due to lack of lysine residues that are optimally positioned to be labelled by BirA fused to Mad1.

Given the strong biotinylation of Bub1 in the Mad1-BirA cell line we tested whether we could detect a direct interaction between Bub1 and the C terminus of Mad1. We purified FLAG-Bub1 146–553/Bub3 and FLAG-Bub1 146–553 ΔCD1/Bub3 complexes from HEK293 cells and Strep-Mad1 597–718 from bacteria, which forms a stable Mad1 dimer[55]. We determined the exact molecular mass of all recombinant Mad1 complexes used in this study by multi-angle light scattering analysis, which confirmed their predicted composition. This Mad1 C-terminal fragment was incubated with Bub1 146–553/Bub3 complexes before being captured by a Strep affinity resin and following washing and elution the binding of

Bub1 was analysed by blotting for FLAG. We could detect binding of FLAG-Bub1 146–553/Bub3 to Mad1 while the removal of ΔCD1 prevented this interaction (Fig. 3d,e).

To further investigate a possible direct interaction between Mad1 and CD1, we used surface plasmon resonance technology. We immobilized a biotinylated CD1 peptide (residues 449–480) as well as a control CD1 peptide with mutations of residues 461–465 to Ala on the surface of the chip and tested the binding to Mad1 597–718 (Fig. 4; Supplementary Fig. 6). To achieve a more efficient synthesis of CD1 peptides and in particularly of the phosphorylated peptides we used norleucine (Nle) instead of methionine (Met) but we cannot exclude that Mad1 will have a different affinity for peptides containing methionine. We could only detect a weak interaction to the unphosphorylated CD1 peptide ($K_d = 304$ μM) and no binding to the control peptide. Given that Ser459 and Thr461 phosphorylation was critical for Bub1 functionality *in vivo* we tested if they affected Mad1 597–718 binding. A CD1 peptide with Thr461 phosphorylated showed a 10-fold stronger binding to Mad1 with a $K_d$ of 32 μM, while phosphorylation of S459 did not stimulate the interaction (Fig. 4a). When both Ser459 and Thr461 where phosphorylated

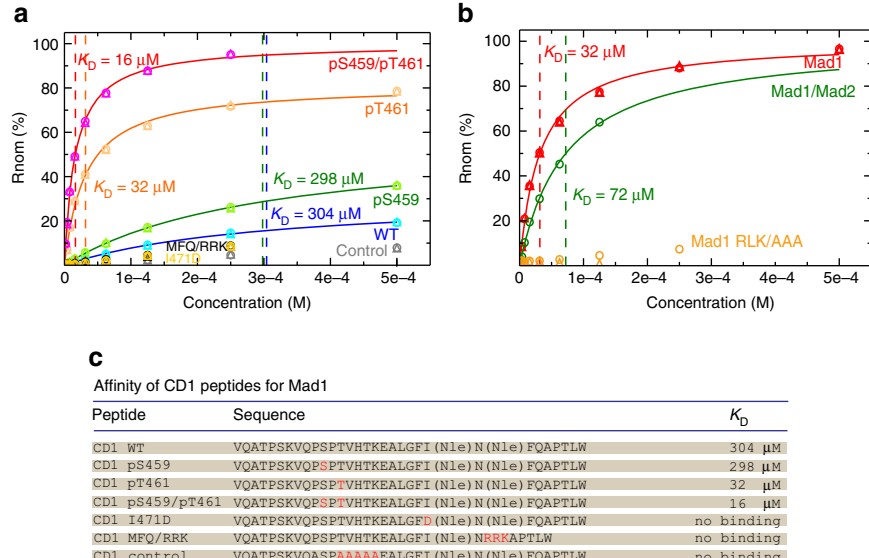

**Figure 4 | Direct binding of Mad1 to CD1 peptide.** (**a**) SPR steady-state responses for control, WT, pS459, pT461, pS459/pT461, I471D, MFQ/RKK and control Bub1 CD1 peptides to purified Mad1 597–718 protein. The biotinylated peptides were captured on a streptavidin-coated CM5 chip and increasing concentrations of Mad1 were subsequently added for binding assessment. Empty symbols represent experimental data points from two technical replicates using the same chip. An additional independent experiment with two replicates has been performed giving in all cases consistent results. The equilibrium dissociation constants, $K_D$, were calculated using a steady-state affinity model and the lines represent the best fit to the model. The experimental data points for the interaction between the control peptide and Mad1 (grey symbols) show not enough curvature to be fitted properly. (**b**) Similar to **a**, SPR steady-state responses for pT461 peptide with purified Mad1 597–718 or Mad1 597–718 RLK/AAA or the Mad1 485–718/Mad2 complex. (**c**) Sequence and summary of the affinity of CD1 peptides for Mad1 597–718 as determined by SPR.

we measured a $K_d$ of 16 μM showing that Thr461 is the phosphorylation site contributing the most to Mad1 binding. A recombinant tetrameric Mad1(485–718)/Mad2 complex also bound to the CD1 peptide phosphorylated on Thr461 with a $K_d$ of 72 μM (Fig. 4b). As the Mad1 RLK motif is required for binding Bub1 in budding yeast *in vivo*[39], we tested whether this motif was required for direct binding to CD1 phosphorylated on Thr461. Mad1 597–718 RLK/AAA migrated at the same position on a size exclusion column as the wild-type protein arguing that the mutations did not affect folding. However, Mad1 597–718 RLK/AAA showed no binding to the phosphorylated CD1 peptide (Fig. 4b). We furthermore analysed the binding of Mad1 597–718 to CD1 peptides harboring the Ile471Asp and MFQ/RRK (residues 474–476) mutations with Thr461 being phosphorylated. Both mutations blocked the binding of Mad1 to CD1 showing that these hydrophobic residues are important for the interaction (Fig. 4a,c).

These results reveal a direct interaction between the C terminus of human Mad1 and the CD1 of Bub1. The interaction is stimulated by phosphorylation of Bub1 on Thr461 and requires the RLK motif of Mad1.

**Fusion of Mad1 and Bub1 bypasses the requirement for CD1.** In live cell experiments we observed that Venus-Mad1 RLK/AAA failed to localize to kinetochores supporting the notion that a direct Bub1-Mad1 interaction facilitates Mad1/Mad2 kinetochore localization in human cells (Supplementary Fig. 7A). In addition to localizing Mad1/Mad2 to kinetochores we considered the possibility that Bub1 CD1 might have additional functions in the checkpoint. This possibility was suggested by observations in fission yeast where the CD1 domain is still required for the checkpoint when Mad1 is tethered to kinetochores[45]. To test this, we fused Mad1 residues 485–718, a region of Mad1 that has previously been shown to support SAC signalling *in vivo*[56], to the N terminus of Bub1 1–553 ΔCD1 (fusion protein referred to as

Mad1—Bub1ΔCD1). All fusion constructs used here have a flexible six amino-acid linker (GSGSGS) between the two proteins. In our Bub1 RNAi system Mad1—Bub1ΔCD1 fully restored SAC signalling (Fig. 5a; Supplementary Fig. 7B). If we mutated the Bub3 binding domain in this fusion protein it no longer localized to kinetochores and failed to support SAC signalling (Fig. 5a; Supplementary Fig. 7B; note that this construct has a negative effect on the checkpoint likely due to Mad1 sequestering soluble Mad2). If we fused Mad2 to the C terminus of Bub1 1–553 ΔCD1 (Bub1 ΔCD1-Mad2) this also bypassed the requirement for CD1 in our Bub1 RNAi assay (Fig. 5b; Supplementary Fig. 8A). Consistently, we could detect Mad1 in immunopurifications of the Bub1 ΔCD1-Mad2 fusion protein arguing that Mad1 is indirectly recruited via Mad2 in this fusion protein (Supplementary Fig. 8B). The ability of Bub1ΔCD1-Mad2 to support SAC signalling in the absence of endogenous Bub1 required that the dimerization interface of the tethered Mad2 was intact because mutating Arg133 and Gln134 to glutamic acid and alanine (Mad2 RQ/EA) resulted in an inactive construct (Fig. 5b). This is consistent with the Mad2 template model for Mad2 activation[57].

Surprisingly, fusion of Mad1 485–718 to amino acids 1–280 of Bub1 (Mad1–Bub1 1–280) was sufficient to support checkpoint activity when endogenous Bub1 was depleted (Fig. 5a). Bub1 1–280 contains the kinetochore targeting part of Bub1 in form of the TPR domain and the Bub3 binding region and the ability of Mad1–Bub1 1–280 to rescue Bub1 depletion argues that the main function of Bub1 in SAC signalling is to properly localize Mad1 at kinetochores.

To rule out that artificial localization of Mad1 to KNL1 bypassed the requirement for Bub1 in the checkpoint we fused Mad1 485–718 to a minimal KNL1 protein encompassing four MELT repeats (amino acids 1000–1200) and the kinetochore-targeting domain of KNL1 (amino acids 1834–2316). Although this construct enhanced SAC signalling in nocodazole-arrested cells this was fully dependent on Bub1 as Bub1 depletion

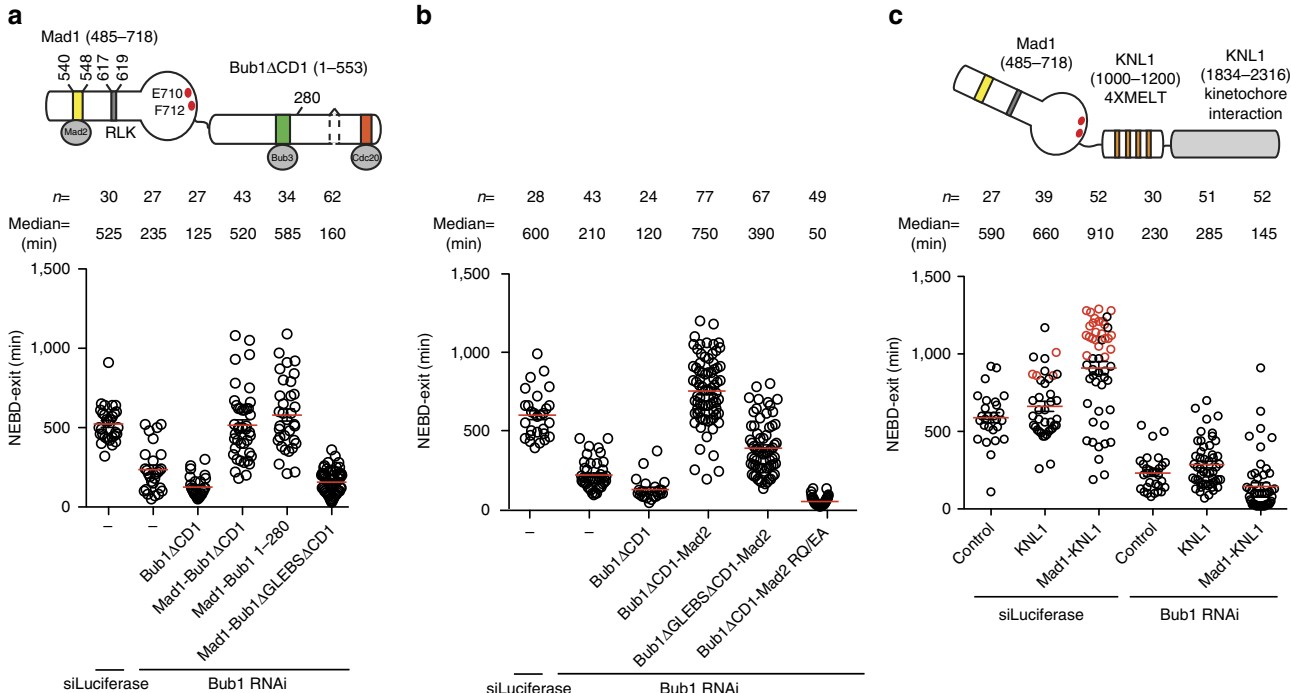

**Figure 5 | Fusing Mad1 or Mad2 to Bub1 bypasses the requirement for CD1.** (**a**) Schematic of the Mad1–Bub1 fusion protein with motifs or amino acids analysed indicated (top). HeLa cells were transfected with a luciferase RNAi oligo or a Bub1 RNAi oligo together with the indicated Venus tagged constructs as outlined in Fig. 1b. Live cell imaging was performed in a low dose of nocodazole and time from NEBD to exit was measured in single cells. Note, Mad1–Bub1 1–280 localizes to kinetochores only in a fraction of cells, which were counted in this study. (**b**) HeLa cells were transfected with Bub1 RNAi oligos together with Venus tagged Bub1 1–553ΔCD1-Mad2 fusion constructs. Mad2 R133E/Q134A (RQ/EA) mutant cannot dimerize with Mad2. Live cell imaging was performed as in **a**. (**c**) Schematic of Mad1 fusion to a KNL1 protein containing four MELT repeats and the kinetochore targeting domain (top). HeLa cells were transfected with (1000–1200)-(1834–2316) KNL1-Venus or Mad1-(1000–1200)-(1834–2316) KNL1-Venus contructs as indicated and either treated with a luciferase or Bub1 RNAi oligo. Live cell imaging and quantification were performed as in **a**. Each circle represents single cell mitotic timing and the red line indicates the median, which is also indicated above. The number of cells analysed per condition is indicated above. Red circles indicate cells still arrested in mitosis at the end of the experiment. The data are from at least two independent experiments.

significantly reduced the time cells spent in mitosis (Fig. 5c; Supplementary Fig. 9). Thus, the ability of Mad1–Bub1 1–280 to replace Bub1 function is not simply a result of localizing Mad1 to KNL1.

From this we hypothesize that Mad1 has to be positioned accurately within the kinetochore to stimulate SAC signalling and that Bub1 through its CD1 brings Mad1 close to KNL1 MELT repeats a function recapitulated by the minimal Mad1–Bub1 1–280.

**Mad1 has additional functions in the SAC.** The lab of Silke Hauf and ours previously reported that Mad1 has a direct role in the checkpoint independently from recruiting Mad2 to kinetochores. This function depends on conserved residues in the C terminus of Mad1 namely Glu710, Phe712 and Arg714 (refs 45,52). We previously speculated that these Mad1 residues could be binding Bub1 and thus be required for Mad1 to be fully functional in SAC signalling.

To investigate this, we first tested the role of these Mad1 residues in binding Bub1. We measured the ability of recombinant Mad1 (597–718) Glu710Ala/Phe712Ala to bind the CD1 peptide phosphorylated on Thr461. This revealed that the Mad1 mutant had the same affinity towards phosphorylated CD1 as wild type Mad1 (597–718; Fig. 6a). Consistently, if we mutated Glu710, Phe712 and Arg714 to Ala in Mad1-BirA (MAD1 EFR/AAA) this did not affect the biotinylation of Bub1 and KNL1, further supporting that these residues are not required for Bub1 binding (Fig. 6b; Supplementary Fig. 10). However, when we mutated Glu710 and Phe712 to Ala in the Mad1–Bub1ΔCD1

fusion we found that this mutant was less efficient in supporting SAC signalling (Fig. 6c; Supplementary Fig. 7). This argues that Mad1 has an important function in the checkpoint in addition to interacting with Mad2 and Bub1.

## Discussion

The kinetochore and checkpoint proteins form a complex set of interactions that facilitate the generation of the 'wait anaphase' signal in the form of the MCC. Understanding SAC signalling requires the elucidation of the molecular interactions between checkpoint components. Here we present experimental data, which unequivocally show a direct Mad1–Bub1 interaction in human cells in agreement with recent studies showing an Mps1 stimulated interaction between the human checkpoint proteins[40,41]. The Mad1 RLK/AAA mutant is unable to bind directly to Bub1 and fails to localize efficiently to kinetochores, arguing that a direct Bub1-Mad1 interaction is important for Mad1 kinetochore localization. This does not exclude contribution from other components such as the RZZ complex that could stabilize the Mad1–Bub1 interaction. Our results suggest that Bub1 has an important role in positioning the Mad1/Mad2 complex in proximity of MELT repeats on KNL1 and we favor that this is to bring it into proximity of Cdc20 and BubR1 to promote MCC formation (Fig. 6d). We propose this model based on a number of observations. First, fusion of the kinetochore-targeting region of Bub1 (Bub1 1–280) to Mad1 generates a minimal construct that supports SAC signalling when endogenous Bub1 is depleted. Second, we find that the fusion of Mad1 to KNL1 is not sufficient to bypass the requirement for

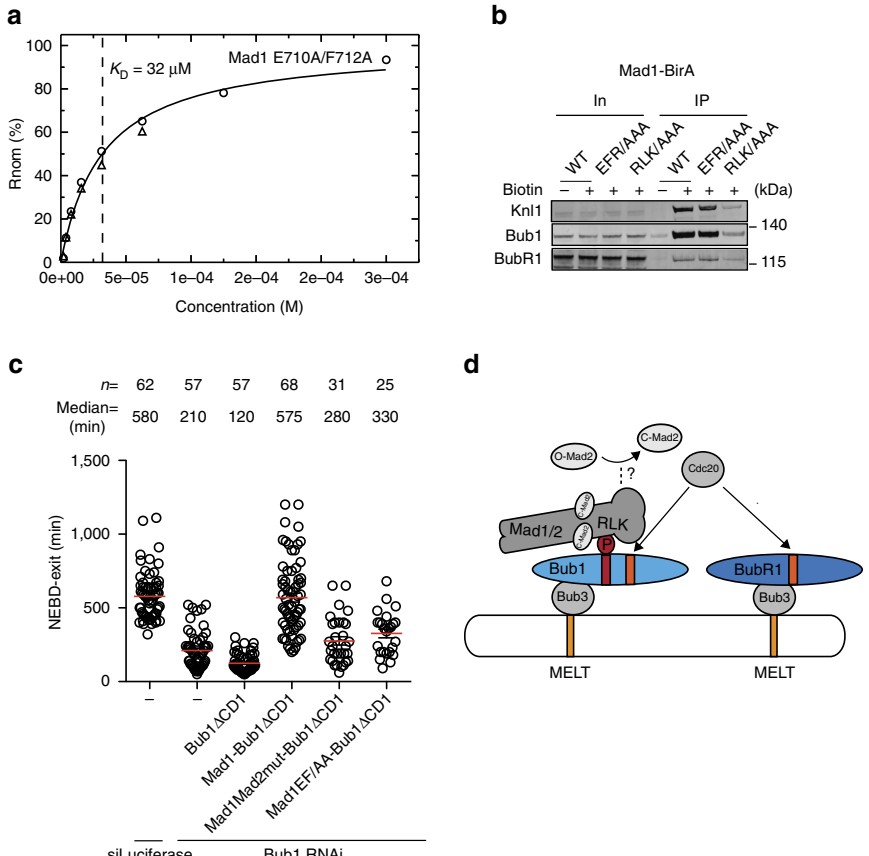

**Figure 6 | Residues in Mad1 are important for SAC function but not Bub1 binding.** (**a**) SPR steady-state responses for pT461 Bub1 CD1 peptides to purified Mad1 597–718 Glu710Ala/Phe712Ala protein similar to Fig. 4. (**b**) Nocodazole arrested stable HeLa cell lines expressing Mad1-BirA or mutants were treated with biotin as indicated and biotinylated proteins were isolated. The samples were analysed for the presence of KNL1, Bub1 and BubR1 by western blot. Mad1-BirA (WT), Mad1(EFR/AAA)-BirA and Mad1 (RLK/AAA)-BirA. Representative of two independent experiments. (**c**) HeLa cells were transfected with Bub1 RNAi oligos together with Venus-Bub1 1–553ΔCD1 or Venus-Mad1–Bub1 ΔCD1 or mutants thereof. The Mad1Mad2mut has its Mad2 binding site mutated such that it cannot bind Mad2. The Mad1 EF/AA mutant has E710A/F712A mutations. Live cell imaging and quantification were performed as in Fig. 5a. Each circle represents single cell mitotic timing and the red line indicates the median, which is also indicated above. The number of cells analysed per condition is indicated above. The data are from at least 2 independent experiments. (**d**) Model of how binding of Mad1/Mad2 to CD1 in Bub1 would position the complex in proximity to the MCC components Mad2, Cdc20 and BubR1. Cdc20 can interact with both the ABBA motif (orange) in Bub1 and the C-terminal ABBA motif in BubR1. The dotted line from the Mad1 C-terminal domain indicates that this domain might facilitate O-Mad2 to C-Mad2 conversion.

Bub1 and in agreement with this, the CD1 is still required in fission yeast when Mad1 is tethered to kinetochores[45].

An interesting aspect uncovered here is that the CD1 needs to be phosphorylated to function in line with observations from budding and fission yeast[37,45,58]. We show that this occurs at kinetochores in prometaphase and our *in vitro* phosphorylation assays implicate Cdk1 and Mps1 as possible kinases. Recent work from the Yu lab suggests that Cdk1 phosphorylates Ser459 to allow subsequent phosphorylation of Thr461 by Mps1 (ref. 40). We have not extensively explored this and our data does not exclude this idea, but we also see that Mps1 can directly phosphorylate Thr461 *in vitro* as observed in budding yeast[37]. Our *in vitro* phosphorylation assays with Cdk1-Cyclin B1 suggest an alternative possibility namely that Cdk1 phosphorylates both sites and it has been shown that Cdk1 can phosphorylate non-Ser/Thr-Pro sites. Indeed, Thr461 has surrounding residues that are favourable for non-Ser/Thr-Pro sites including a Lysine residue in the +4 position[59].

We show that the function of Thr461 phosphorylation is to facilitate Mad1 binding, which is fully consistent with recent data from the Yu lab[40] and the work from budding yeast[37]. Elegant

work using budding yeast proteins revealed that a large internal Bub1 fragment phosphorylated by Mps1 can bind Mad1/Mad2 directly, but the exact region of Bub1 and which phosphorylation sites bind to Mad1 were not identified[37]. Our result and that of the Yu lab support that the key contact site of budding yeast Mad1 is likely CD1, explaining its conservation[37,39,45,46]. We note that worm Bub1 has lost CD1 potentially explaining why Mad1 binds the kinase domain.

In addition to the CD1 phosphorylation sites, hydrophobic residues that might constitute part of an alpha-helical segment of CD1 are also critical for Bub1 function and Mad1 binding. An interesting idea is that the Thr461 phosphorylation binds to the RLK motif and the hydrophobic residues from the alpha helical segment of CD1 interacts with the coiled-coil part of Mad1.

Another important function of Bub1 is to localize a fraction of Cdc20 to kinetochores through its ABBA motif[31,32]. In the Mad1–Bub1 fusions we do not see a requirement for the ABBA motif, while we observe this in Bub1 alone. This is likely due to a redundancy of the ABBA motif in Bub1 and the C-terminal ABBA motif in BubR1 and that both contribute to localization of Cdc20 to kinetochores[31,34]. Since Mad1 binds CD1, it would

position the Mad1/Mad2 complex close to Cdc20 bound to the ABBA motif in Bub1, hereby facilitating Mad2-Cdc20 complex formation and recent *in vitro* reconstitution data support this[40,41]. This would explain why removal of the ABBA motif from Bub1 results in reduced checkpoint activity. However, the ABBA motif of Bub1 is not as critical for Bub1 functionality as the CD1 and we favour this is due to BubR1 recruiting Cdc20 as well.

Finally, this work provides novel insight into our previous observations of a direct role of Mad1 in the checkpoint[52]. Since mutations in the C-terminal globular part of Mad1 are still critical for Mad1 functionality in the Mad1–Bub1 fusion, this argues that these residues are important for an aspect of Mad1 function unrelated to Bub1 binding. Consistent with this is the recent *in vitro* reconstitutions of MCC formation showed that the Mad1/Mad2 complex acts catalytically when phosphorylated by Mps1 even in the absence of Bub1 (refs 40,41). Interestingly, Mps1 phosphorylates Ser713 and Thr716, which are close to Glu710 and Phe712 that we have mutated. One possibility is that these mutations prevents binding of Cdc20 to the Mad1 C terminus[40].

In conclusion, we dissect a direct phosphoregulated interaction between Bub1 and Mad1 and uncover its role in the checkpoint in human cells.

## Methods

**Cell culture techniques.** HeLa cells (ATCC) were cultivated in DMEM medium (Invitrogen) supplemented with 10% fetal bovine serum and antibiotics. Cells were synchronized by thymidine (2.5 mM) the day before co-transfection with siRNA oligos against Bub1 (100 nM as final concentration) and corresponding plasmids by Lipofectamine 2000 (Life Technologies). RNAi oligos targeting Bub1 (5′-GAGU-GAUCACGAUUUCUAA-3′) or luciferase (5′-CGUACGCGGAAUACUUCGA-3′), were used for RNAi depletions. Thymidine was added again 12 h later after transfection. A second RNAi was performed during the second thymidine block. Twenty four hours later, the cells were released from thymidine block. Filming or fixation was performed when cells entered mitosis.

For SILAC labelling, HeLa cells were grown in SILAC DMEM medium (Invitrogen) supplemented with 10% dialysed FBS, L-glutamine, penicillin/streptomycin and isotope labelled arginine and lysine. L-lysine and L-arginine (Sigma) were used for light culture. L-lysine 4,4,5,5-D4 and L-arginine–U-13C6, or L-lysine-U-13C6-15N2 and L-arginine–U-13C6-15N4 (Cambridge Isotope Laboratories) were used for medium or heavy culture.

**Cloning.** Venus-Bub1 1–553 and all its variants were cloned into N-Venus pcDNA5/FRT/TO plasmid with KpnI and NotI sites on 5′ and 3′ ends. Venus- Mad1–Bub1 1–553 ΔCD1 was cloned by inserting Mad1 (cDNA coding for 485-718aa) into Venus-Bub1 at KpnI site while Venus-Bub1 1–553ΔCD1-Mad2 was cloned by inserting full length Mad2 using NotI. pGEX Bub1 425–500 was cloned by amplifying this fragment of Bub1 by PCR and cloning it into BamHI/NotI of pGEX. Full-length Mad1 was PCR amplified and cloned into pcDNA5/FRT/TO-BirA C-term using KpnI and BamHI or pcDNA5/FRT/TO-BirA N-term using BamHI and NotI to generate Mad1-BirA or BirA-Mad1 fusion constructs. Indicated point mutations were introduced into Mad1 using a QuickChange PCR mutagenesis strategy. Details of cloning will be provided upon request.

**Purification of biotinylated protein complexes.** Stable HeLa cell lines expressing the indicated Mad1 BirA fusion proteins were exposed to 0.1 ng ml$^{-1}$ doxycycline for 18 h to obtain near endogenous Mad1 expression levels. Cells were arrested in mitosis by a single thymidine block and subsequent nocodazole (150 ng ml$^{-1}$) treatment for 16 h. Biotinylation of proximity interactors was induced by the addition of a final concentration of 25 μM of biotin simultaneously with the addition of nocodazole. Mitotic cells were collected and washed three times in PBS before lysed in RIPA buffer (50 mM Tris pH 7.5, 150 mM NaCl, 1 mM EDTA, 1% Nonidet P-40, 0.25% Na-deoxycholate, 0.1% SDS) containing protease inhibitors (Roche). Cell lysate was clarified by centrifugation and incubated overnight at 4 °C with High Capacity Streptavidin Resin (Thermo Scientific). Streptavidin beads were washed once with RIPA buffer followed by two washes with water containing 2% SDS and a final wash with RIPA buffer. Biotinylated proteins were eluted from the streptavidin beads with 2× Laemmli LDS sample buffer containing 1 mM of biotin before separated on 4–12% Bis-Tris NuPage gels (Life technologies). After separation, proteins were either examined by western blot or processed for quantitative SILAC mass spectrometry (see below).

**Mass spectrometric analysis.** Cells were collected and biotinylated protein complexes purified as described above. The indicated LDS sample buffer eluates were combined and resolved by SDS–PAGE gel electrophoresis. The Coomassie-stained gel was cut into slices, reduced with 1 mM DTT, alkylated with 5 mM chloroacetamide, digested using modified sequencing grade trypsin (Sigma) and loaded on to C18 stage tips before mass spectrometric analysis.

All mass spectrometry experiments were performed on a nanoscale EASY-nLC 1000 UHPLC system (Thermo Fisher Scientific) connected to an Orbitrap Q-Exactive Plus equipped with a nanoelectrospray source (Thermo Fisher Scientific). Each sample was eluted off the StageTip, auto-sampled and separated on a 15 cm analytical column (75 μm inner diameter) in-house packed with 1.9-μm C18 beads (Reprosil Pur-AQ, Dr Maisch) using a 3 h gradient ranging from 5 to 64% acetonitrile in 0.5% formic acid at a flow rate of 200 nl min$^{-1}$. The effluent from the HPLC was directly electrosprayed into the mass spectrometer. The Q Exactive Plus mass spectrometer was operated using data-dependent acquisition, with all samples being analysed using a 'sensitive' acquisition method and a normalized collision energy (NCE) of 28. Back-bone fragmentation of eluting peptide species was obtained using higher-energy collisional dissociation (HCD), which ensured high-mass accuracy on both precursor and fragment ions. All raw data analysis was performed with MaxQuant software suite version 1.3.0.5 supported by the Andromeda search engine. For SILAC quantification a minimum of two ratio-counts was required.

A similar set-up was used for analysing the phosphorylated GST-Bub1 except that an EASY-nLC 1200 UHPLC system connected to an Orbitrap Q-Exactive HF was used and the acetonitrile gradient was run for 77 min. The samples were analysed using a 'fast' acquisition mode with a normalized collision energy (NCE) of 25 and raw data analysed with MaxQuant software suit version 1.5.7.4.

**Circular dichroism.** Circular dichroism spectra were recorded at 20 °C in a Jasco J-815 spectropolarimeter using a 0.2 cm path length quartz cell cuvette, 260–190 nm measurement range, 50 nm min$^{-1}$ scanning speed, 2 s response time and 0.5 nm data pitch. Peptides at 30 μM were dissolved in water or in the organic solvent 2,2,2-trifluoroethanol.

**Surface plasmon resonance.** Bub1 peptides were purchased from Biosyntan Gmbh (Germany). The purity obtained in the synthesis was 95–98% as determined by high-performance liquid chromatography (HPLC) and the identity of the peptides was confirmed by mass spectrometry. SPR experiments were performed on a Biacore T200 (GE Healthcare) at 25 °C. Streptavidin (ThermoFisher Scientific) at 0.05 mg ml$^{-1}$ in 10 mM sodium acetate, pH 4.5, was directly immobilized on a CM5 sensor chip (GE Healthcare) by standard amine coupling chemistry with N-hydroxysuccinimide and N-ethly-N′-(3-dimethlyaminopropyl)carbodiimide up to an immobilization level of approximately 2000 RU. Unreacted sites were quenched with ethanolamine HCl. Subsequently, N-terminal biotinylated Bub1 peptides (at 10 nM in HBS-EP + buffer and flow rate 5 μl min$^{-1}$) were captured at ∼50 RU, respectively, in channels 2, 3 and 4 of the streptavidin coated chip, leaving channel 1 free for reference. An identical approach was used to prepare a second CM5 chip with the negative controls. In this case, a N-terminal biotinylated 5A mutant, a non-related phosphorylated peptide and the Bub1 pT461 peptide were captured, respectively, in channels 2, 3 and 4. HBS-EP + (10 mM Hepes, pH 7.4, 150 mM NaCl, 3 mM EDTA, 0.05% (v/v) Surfactant P20) was used as running buffer for both streptavidin immobilization and peptide capture. Before the affinity measurements, both Mad1 and Mad1 RLK/AAA were extensively dialysed against the assay running buffer 25 mM Tris, pH 7.8, 150 mM NaCl, 10 mM MgCl₂, 0.25 M TCEP, 0.005% (v/v) Surfactant P-20. Mad1 and the Mad1 RLK/AAA mutant were injected (at a flow rate of 10 μl min$^{-1}$) over the sensor surfaces at eight concentrations (starting at the lowest concentration of 3.9 μM (Mad1) or 1.95 μM (RLK)) prepared by twofold serial dilution in assay buffer of a 500 μM (Mad1) or 250 μM (RLK) stock. The dissociation time was set-up at 180 s. Sensorgrams were analysed after double referencing. Equilibrium was reached over the course of the injection and the average RU values for each protein concentration at steady state (5 s before the end of the injection) plotted against protein concentration and fitted by nonlinear regression to a steady-state affinity model to determine the dissociation constants, $K_D$s.

**Protein expression.** Purified FLAG Bub1 146–553/Bub3 was obtained from HEK293 cells by transfecting ten 15 cm dishes, $1 \times 10^7$ cells per dish, each with 15 μg FLAG Bub1 146–553 and 15 μg untagged Bub3 plasmids. Two days after transfection cells were collected and FLAG Bub1 146–553/Bub3 purified by lysing cells in buffer L (350 mM NaCl, 50 mM Tris pH 8.0, 0.05% NP40) and following centrifugation the lysate was incubated with 300 μl anti-FLAG beads (Sigma). Following washing with buffer L the Bub1 146–553/Bub3 complex was eluted with buffer L containing 200 ng μl$^{-1}$ FLAG peptide by incubating for 20 min. Bub1 146–553/Bub3 was stored at 4 °C.

His-Strep tagged Mad1(597–718) and His tagged Mad1(485–718)/Mad2 protein was expressed in BL21(DE3) cells at 18 °C overnight with 0.5 mM IPTG in the medium. Following lysis the protein was purified on HiTrap Ni column and eluted in buffer E (50 mM NaP pH 7.5, 300 mM NaCl, 500 mM immidazole, 10% glycerol, 0.5 mM TCEP) and peak fractions were further purified on HiLoad 16/60

superdex 75 column equilibrated with buffer G (50 mM NaP pH 7.5, 150 mM NaCl, 10% glycerol, 0.5 mM TCEP). Before the superdex 75 column the Mad1 (485–718)/Mad2 complex was run on a Mono Q 10/100 GL column and peak fractions pooled for size exclusion. Peak fractions were aliquoted and stored at −80 °C.

GST- Bub1 (425–500) was expressed in BL21(DE3) cells at 18 °C overnight with 0.5 mM IPTG in the medium. The bacteria were lysed in 50 mM Tris-HCl pH 7.4, 300 mM NaCl, 10% glycerol, 5 mM β-mercaptoethanol, 1 mM PMSF, plus Complete-EDTA free protease inhibitors (Roche) using a high pressure homogenizer. Lysates were cleared by centrifugation and incubated with pre-equilibrated Glutathione-Fast Flow (GE Healthcare) beads for 2 h at 4 °C and eluted with GST-elution buffer at room temperature (50 mM Tris pH 8.8, 300 mM NaCl, 10% glycerol, 5 mM β-mercaptoethanol, 20 mM reduced glutathione). Elutions were pooled and run on Superdex 200 column (GE Healthcare) in PBS with 8.7% glycerol and 5 mM mercaptoethanol. Peak fractions were aliquoted and stored at −80 °C.

**Immunofluorescence and quantification.** Cells growing on coverslips were synchronized with a double thymidine block and RO3306 (10 nM) block (for Cdc20 staining only). After washing with PBS, the cells were treated with medium containing nocodazole (200 ng ml$^{-1}$) for 1–2 h and fixed with 4% paraformaldehyde in PHEM buffer (60 mM PIPES, 25 mM HEPES, pH 6.9, 10 mM EGTA, 4 mM MgSO$_4$) for 20 min at room temperature. Fixed cells were extracted with 0.5% Triton X-100 in PHEM buffer for 10 min. The antibodies used for cell staining include Bub1 (Abcam, ab54893, 1:400), Cdc20 (Millipore, MAB3775, 1:200), CREST (Antibodies Incorporated, 15–234, 1:400) and GFP (Abcam, ab290, 1:400). Streptavidin-FITC (Sigma S3762, 1:200) was used for staining of BirA-Mad1 or Mad1-BirA cells. The Bub1 phospho antibody was raised against a 50:50 mix of C-Ahx-PSKVQP[pS]P[pT]VHTKEA-amide and Acetyl-SKVQP[pS]P[pT]VHTKEA-Ahx-C-amide peptides coupled to carrier and the antibody was affinity purified (21st Century Biochemicals) and used at a 1:400 dilution. All the fluorescent secondary antibodies are Alexa Fluor Dyes (Invitrogen, 1:1,000). Z-stacks 200 nm apart were recorded on a Deltavision Elite microscope (GE Healthcare) using a × 100 oil objective followed by deconvolution using Softworx before quantification. Protein intensity on kinetochores was quantified by drawing a circle closely along the rod-like CREST staining covering the interested outer kinetochore protein staining on both ends. The intensity values from the peak three continuous stacks were subtracted of the background from neighbouring areas and averaged. The combined intensity was normalized against the combined CREST fluorescent intensity.

**Immunoprecipitation and in vitro protein binding assay.** HeLa cells in 15 cm dishes were transfected with either luciferase or Bub1 RNAi oligos twice or corresponding constructs once within 48 h. A volume of 200 ng ml$^{-1}$ of nocodazole was used to arrest cells before the mitotic cells were collected and lysed in lysis buffer containing 10 mM Tris pH 7.4, 150 mM NaCl, 0.5 mM EDTA and 0.5% NP-40 with both protease and phosphatase inhibitors (Roche). After centrifugation at 17,000g for 10 min, the supernatant was applied to 20 μl of rec-Protein G sepharose beads (Thermo Fisher) beads pre-crosslinked with a pan Mad2 antibody[60] or GFP Trap beads (Chromotek) and shaken at 4 °C for 2 h. After three times of washing with 0.5 ml lysis buffer, the bound protein was washed by boiling in 50 μl 2 × LDS loading buffer. Western blot was performed afterwards.

For binding experiments between Mad1 and Bub1, 8 μg of His-Strep-Mad1 597–718 was incubated with 1 μg FLAG-Bub1 146–553/Bub3 complexes in 125 μl of binding buffer (30 mM Tris pH 7.4, 50 mM NaCl, 0.1 mM DTT) for 30 min at 30 °C. The reaction was added to 20 μl Strep-Tactin beads (GE Healthcare) and incubated an additional 45 min. The beads were washed three times with 500 μl wash buffer (30 mM Tris pH 7.4, 120 mM NaCl, 0.1 mM DTT, 0.1% NP-40) and then complexes eluted with 60 μl elution buffer (100 mM Tris pH 8.0, 120 mM NaCl, 2.5 mM desthiobiotin) for 15 min. The samples were analysed by SDS–PAGE and stained with coomasie or analysed by western blot.

The following antibodies were used for western blot: BubR1 (Rabbit polyclonal, 1:5,000, produced in house), Cdc20 (sc-13162, 1:1,000, Santa Cruz), FLAG (F3165, 1:2,000, Sigma), Ndc80 (ab3613, 1:1,000, abcam), Mad2 (A300-301A, 1:1,000, Bethyl Laboratories), Bub1 (ab54893, 1:1,000, abcam), Bub3 (611730, 1:1,000, BD Biosciences), Knl1 (Rabbit polyclonal, 1:500, produced in house), APC7 (A302-550A, 1:2000, Bethyl Laboratories).

***In vitro* phosphorylation of GST Bub1.** Recombinant GST-Bub1 (12 μg) was incubated with 20 units CDK1-CyclinB1 (New England Biolabs), 0,2 μg Mps1 (TTK; Life Technologies) in protein kinase buffer (50 mM Tris pH 7.5, 10 mM MgCl$_2$, 0.1 mM EDTA, 2 mM DTT, 0.01% Brij 35). 500 μM ATP and 125 nM Calyculin A (Cell Signaling) were added and the reactions were incubated at 30 °C for 1 h. Controls were incubated in kinase buffer and Calyculin A only. 6 μg of phosphorylated protein was resolved on NuPAGE Novex 4–12% Bis-Tris protein gels, which were fixed and stained by Colloidal Blue Staining (Thermo Scientific). The band corresponding to GST-Bub1 was cut out and analysed by mass spectrometry.

**Live cell imaging.** Live cell imaging was performed on a Deltavision Elite system using a × 40 oil objective (GE Healthcare). Cells transfected with RNAi oligos and corresponding plasmids in a six-well plate were re-seeded in eight-well Ibidi dishes (Ibidi) 1 day before the filming. Growth media was changed to Leibovitźs L-15 (Life technologies) containing low dose nocodazole (30 ng ml$^{-1}$) before filming. Appropriate channels were recorded for 22 h and the data were analysed using Softworx (GE Healthcare). Statistical analysis was done using Prism software. At least three independent experiments were performed for all live cell experiments and the data pooled from multiple experiments are presented in figures.

**Data availability.** All relevant data are available from the authors upon request.

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

## Acknowledgements

We thank H. Yu, N. Davey and E.P.T. Hertz for interesting discussions on the CD1-Mad1 interaction and the CPR protein production facility and in particular A.L. Lino Vala and K. Pardes in the prokaryotic expression team for purifying recombinant Mad1 proteins. We thank A. Musacchio for providing the expression construct for Mad1(485–718)/Mad2. The Novo Nordisk Foundation Center for Protein Research, University of Copenhagen, is supported financially by the Novo Nordisk Foundation (grant agreement NNF14CC0001). In addition, this work was supported by grants from the Danish Cancer Society (R72-A4351-13-S2 and R124-A7827-15-S2) to J.N., The Danish Council for Independent Research (11-105247) to G.Z. and DFF-4181-00340 to J.N. M.L.N was supported by grants from the Novo Nordisk Foundation (grant number NNF13OC0006477); The Danish Council of Independent Research, grant agreement number DFF 4002-00051 (Sapere Aude) and grant agreement number DFF 4183-00322A.

## Author contributions

J.N. and G.Z. planned the project and designed the experiments. G.Z. performed all experiments except Mad1 BioID experiments that were performed by T.K., mass spectrometry that was performed by K.B.S., S.S. and M.L.N., SPR and CD experiments that were performed by B.L.-M. and Bub1 purification and binding experiments that were performed by D.H.G and J.N.

## Additional information

**Competing interests:** The authors declare no competing financial interests.

