## [Peer Review File · Nature Communications]

Reviewers' Comments:

Reviewer #1 (Remarks to the Author)

Zhang et al present a biochemical, biophysical and functional analysis of the interactions between human Bub1 and Mad1 and the functional consequences of this interaction. The authors identify the conserved domain 1 (CD1) in Bub1 that is involved in Mad1 direct binding and demonstrate that this interaction is mediated through the phosphorylation of CD1 on Thr461 as well as the hydrophobic residues (Ile471, Met-Phe-Gln: 474-476) from the predicted alpha helical segment. Furthermore the studies demonstrated that the main function of Bub1 is to position Mad1 close to KNL1 MELT repeats.

Although the Mad1 interaction with Bub1 was already reported (*Genes and Development*; 28, 140, 2014), this manuscript is further elucidating this important protein-protein interaction and underlies molecular mechanisms linked to spindle assembly checkpoint signaling at kinetochores in human cells. The data are interesting and novel, but the authors should revise their manuscript and address several specific points given below.

1. There is inconsistency in the peptides and protein constructs that were tested for the binding and functional studies, which needs a clarification and additional experiments:

- In Fig. 3F and H the authors are testing the binding of CD1 peptides using SPR. In the CD1 WT peptide the Met (Fig. 2A) is substituted with Nle amino acid. The authors are not commenting why this substitution was done. Usually Nle is used to probe the role of methionine, but it is important to be determined what the binding affinity of the WT is in order to compare and draw conclusions. This result is missing in the paper.

- In addition in Fig. 2B it is showed that four hydrophobic residues I471 and MFQ (474-476) are important for the checkpoint activity. It will be important to be tested if these mutations are also affecting the direct binding of Bub1 to Mad1 which will reveal more insight of the CD1/Mad1 interaction and checkpoint activity. Thus authors should perform binding studies with peptides where these hydrophobic residues are mutated.

- Although using peptides are useful tools for mapping the binding site of the direct interactions and together with mutagenesis studies is allowing characterization of the interactions nature, it is important to be demonstrated that recombinant Bub1 protein has similar binding affinity as the peptide segment. These results will further provide evidence that the CD1 domain is the interacting site.

- For the binding study the authors are using Mad1 construct (597-718) which is 112 residues shorter than the construct used in the follow up functional assays (Mad1 485-718). The authors need to provide explanation why different constructs were used and evidence that they retain similar kinetochore-binding function.

2. This study is conclusively demonstrating that the direct Mad1-CD1 interaction is stimulated by phosphorylation and furthermore identifies the residues that are phosphorylated and important for the interaction. It was already reported that this interaction is mediated by phosphorylation and authors are discussing this and cite relevant references. Based on the reported studies, in the discussion authors are proposing Cdk1 and Mps1 as potential kinase candidates. Given the quality of the data and to further advance the findings on the phosphorylation role, the authors should extend their studies and use commercially available small-molecule inhibitors of these kinases as chemical tools to test and validate Cdk1 and/or Mps1 involvement in the phosphorylation of CD1 domain. This would provide important insights on the Mad1-CD1 interaction as well as towards elucidating the role of these kinases in regulating the spindle assembly checkpoint signaling. This will increase the novelty of the manuscript and its visibility and interest to researchers in other related disciplines.

Reviewer #2 (Remarks to the Author)

The spindle assembly checkpoint prevents gains or losses of chromosomes during their segregation to the daughter cells in mitosis. This manuscript reports an analysis of the significance of an interaction of two crucial components of the checkpoint, Mad1 and Bub1, in human cells. This interaction has been observed before in other organisms (originally by K. Hardwick's laboratory and later by S. Biggins' in *S. cerevisiae*, in papers cited by the authors). The present work focuses on a segment of Bub1, called the Conserved Domain 1 (CD1), and on the important demonstration that this motif interacts with Mad1. Although two very recent papers have reached partly overlapping conclusions (see below), the authors' clearly developed their observations independently. Their observations are important and significantly advance our understanding of the mechanistic basis of checkpoint signaling. The data are generally adequately supporting the conclusions. Overall, this makes for an interesting contribution to the spindle checkpoint field and I support its publication provided that the following points are adequately considered.

Specific points

The introduction, and in particular several claims of priority, will need to be revised in light of recent publication of two papers demonstrating an interaction of human Mad1-Mad2 and Bub1 (Ji et al. *eLife* 2017; Faesen et al. *Nature* 2017. Partial overlap of these two publications with the work discussed here should not be seen, however, as an obstacle towards publication of this article). We also note that another priority claim in the Introduction, that Bub1 is responsible for the precise positioning of Mad1-Mad2, has already been demonstrated (Krenn et al. *Current Biology* 2014, which the authors cite but in a different context).

Page 6, lines 3-4

The authors should give credit to the work of the scholars who first identified the CD1 and ABBA motifs (Meraldi, Yu, Pines, Kops)

Page 7, line 5

"A likely explanation for the dominant-negative effect..." Here the argument is circular. The authors have not demonstrated a dominant-negative effect (which strictly speaking configures an antagonistic relationship to the wild type allele), but rather a reduction in the duration of mitotic arrest in presence of the deltaCD1 construct that is even more pervasive than after depletion of endogenous Bub1. Whether this is due to a dominant-negative effect is unclear at this stage. To prove it, the authors should demonstrate such an effect in the presence of endogenous Bub1. They may elect to do this experiment, or alternatively refocus the discussion and avoid the confusion that this definition will inevitably arise given that in principle the cells in which these effects are observed ought to have been depleted of endogenous Bub1 (i.e. in cells where no "dominant" effect should be observed). This concern extends to all subsequent instances of "dominant-negative" in the manuscript.

Page 8, lines 12-13

The authors provide no rigorous evidence in this context that the effect they describe is due to lack of phosphorylation of CD1. I agree that this is a reasonable hypothesis, but it is not the only one (e.g. mutations to Ala may be not tolerated). I.e. the authors have not shown a role of phosphorylation, but rather have shown that mutation of two residues that may be phosphorylated provokes the same effect as the deletion of CD1.

Page 9, line 16

Give an angle to that angel

Page 10, lines 1-6

These results are nicely consistent with those published by the Yu laboratory in *eLife* (op. cit.)

Page 10, line 25

When referring to Mad1-Bub1 fusion proteins, the authors should clearly indicate Bub1's name. The term 'Mad1 deltaCD1' is uninformative. Please use Mad1-Bub1(deltaCD1) or equivalent. (Same idea for Mad1-MBR on the next page.)

Page 11, lines 13-14

Residues 1-280 of Bub1 contain more than the MELT Binding Region, and therefore the nomenclature is incorrect. The GLEBS motif of Bub1 is sufficient for kinetochore localization and for MELT binding (work of S. Taylor's and A. Musacchio's laboratories). In the following lines (15-18) it would be fair to cite previous work in support of the authors' statement.

Page 13, line 1

'forms' should be in singular form

Page 13, line 6

'elusive' is no more.

Page 13, lines 23-25

The authors should point out here that the phosphorylation site they have described (pT-461) is clearly embedded in a CDK consensus site (SPTV), and that previous work from the Biggins, Musacchio, and Yu laboratories cited above had collectively implicated both Cdk1 and Mps1 in its creation. This is discussed on page 14, but the discussion could be unified in view of the more recent data.

Page 13, last line

Please remove claims of novelty

Page 15, line 1

In my opinion 'independently' is not the right word, as it implies that these residues can perform a function in the SAC independently of the other motifs and interactions, which is clearly not the case, or at least it is not demonstrated here. I think that the authors mean 'in addition'.

Figure 1F

Error bars missing

Figure 3F

If I understand things properly, the 'control peptide' is actually a mutant, which is indicated differently in the Table. Can the authors harmonize the description of this peptide?

In general, the authors could significantly improve usage of space in their figures, making some panels larger and avoiding large white open spaces.

Reviewer #3 (Remarks to the Author)

The spindle checkpoint is a surveillance mechanism that monitors the chromosome attachment with spindle microtubules during mitosis. Until all sister chromatids are properly attached to microtubules, the checkpoint detects unattached kinetochores and generates diffusible "wait signals" to delay the onset of anaphase. A functional spindle checkpoint is critical for eukaryotic cells to prevent chromosome missegregation and aneuploidy. Bub1 is a multifunctional checkpoint kinase that is essential for the spindle checkpoint activation. Previous studies on the non-kinase functions of Bub1 have led to the discoveries of two important motifs in its middle region, the conserved domain 1 (CD1) and the ABBA motif. The ABBA motif of Bub1 contributes to checkpoint activation by recruiting Cdc20 to kinetochores. In contrast, the function of Bub1 CD1 is less understood in human cells. A recent study (Ji et al., 2017) has reconstituted in vitro the phosphorylation-mediated Bub1 CD1 interaction with Mad1 using purified human proteins.

Mutations that disrupt Mad1-Bub1 interaction in vitro failed to activate the spindle checkpoint in human cells. But the direct evidence showing that the complex of Mad1-Bub1 indeed exists and is functionally important is still lacking. In this manuscript, the authors have further characterized the function of Bub1 CD1 in binding with Mad1 in human cells. The novel discoveries in this study include: (1) the authors provided the in-vivo evidence of the proximity of Mad1 and Bub1, which requires the RLK motif of Mad1; (2) the authors demonstrated that the function of CD1 is to bring Bub1 and Mad1 in proximity, which can be bypassed by Mad1-Bub1 fusion; (3) the authors revealed that an alpha helix of Bub1, C-terminal to the phosphorylated Ser and Thr residues, was also required for Bub1 function, presumably by contributing to Mad1-Bub1 binding.

Overall, this study has presented several interesting findings that could further our knowledge of the checkpoint function of Bub1, even considering the fact that the binding of Mad1 and Bub1 has been reconstituted in the recently published study (Ji et al., 2017). In particular, the in-vivo evidence showing that Mad1 and Bub1 are in proximity using BirA technique and the finding that Mad1-Bub1 fusion protein can bypass the functional requirement of Bub1 CD1 are very significant. Thus publication in Nature Communication is recommended. On the other hand, due to the recently published two studies on the in-vitro reconstitution of spindle checkpoint signaling (Faesen et al., 2017; Ji et al., 2017), the manuscript needs a major revision, highlighting the unique discoveries and discussing the two published studies as appropriate. The authors need to address the following major points in order to consolidate some of the major conclusions that are novel in the manuscript. In addition, several minor points need to be addressed as well prior to publication.

Major points

1. In Figure 3C and Figure 5B, the authors showed that Mad1-BirA fusion protein could biotinylate Knl1, Bub1 and BubR1 in an RLK-dependent manner. Considering the fact that the RLK motif is required to establish the direct interaction between Mad1 and Bub1, these lines of evidence therefore imply that Mad1-BirA binds with Bub1 and biotinylate it. To further confirm this inference, it will be informative to test if the biotinylation can be compromised by Mad1-binding deficient mutants of Bub1, such as S459A and T461A. In addition, it is surprising to see Mad2, which forms a constitutive complex with Mad1, is not biotinylated in Figure 3C. Is there any explanation for this? Moreover, were the Mad1-BirA and BirA-Mad1 proteins themselves biotinylated in the assay?
2. In Figure 2B, the authors have shown that the alpha helix of Bub1 CD1 is required for Bub1 function. But the mechanism of the alpha helix function was not further pursued. Since this alpha helix has not been characterized previously, the authors might want to provide evidence that the alpha helix contributes to the Mad1-Bub1 binding in order to enhance the novelty of the story.
3. In Figure 5, the authors pointed out a direct role of Mad1 that is independent of Bub1 binding. This observation is consistent with the two recently published works (Faesen et al., 2017; Ji et al., 2017). Ji et al. suggest that this Bub1-independent function is through a phosphorylation-dependent direct interaction between Mad1 and Cdc20. The authors should test if Mad1 E710/F712 works together with phosphorylated T716 to bind with Cdc20.

Minor points

1. In Figure 4A, it is surprising to see that Mad1-MBR could fully support the checkpoint activity in the absence of endogenous Bub1. How do the authors rationalize the fact that Mad1-MBR bypasses not only the Mad1-binding function of CD1 but also the Cdc20-binding function of ABBA?
2. It has been showed that the doubly phosphorylated Bub1 CD1 peptide binds with Mad1 stronger than either of the singly phosphorylated Bub1 CD1 peptides (Ji et al., 2017). Thus, the authors should repeat the SPR binding assays in Figure 3F, 3G and 5A with the pS459/pT461 peptide.
3. Figure 5D. There is no evidence that Bub1 CD1 binds to the head of Mad1 C-terminal domain. According to previous studies, it is more likely that Bub1 CD1 directly binds to the RLK motif of Mad1, which is located at the stem of Mad1 C-terminal domain, not the head. Therefore it is better to draw the Mad1-Bub1 binding in a different configuration.
4. In supplementary Figure 5, the Strep-cy2 could detect all kinetochore proteins that were

biotinylated by BirA. It is incorrect for the author to conclude that the BirA fusion proteins localize to kinetochore (Page 34, line 5) based on this data.

5. Page 16, line 18-23. It will be helpful if the authors in this section disclose the amino acid sequences, including the linker regions, of all the fusion proteins used in this study, e.g. Mad1-Bub1, Bub1-Mad2, Mad1-BirA and BirA-Mad1.

6. In the BirA-mediated biotinylation assays, it is necessary to show the expression levels of all BirA fused Mad1 proteins comparing the endogenous Mad1.

7. In Figure 3H, it was not clear to the reader why the authors used Nle to replace methionine.

8. Page 11, line 10. The Mad2 mutations should be Arg133 and Gln134 being mutated to glutamic acid and alanine, respectively.

9. Page 30, line 10. "25 mM biotin" seems to be exceeding biotin solubility in water. It might be a typo of "25 μ M".

10. Page 31, line 7. The panel "H" was mislabeled as panel "G".

11. Page 31, line 18. "R133Q/E134A" is a typo. It should be "R133E/Q134A".

12. Page 32, line 11. "Live cell imaging and quantification were performed as in A)" is referring to the wrong panel.

REFERENCE

Faesen, A.C., Thanasoula, M., Maffini, S., Breit, C., Muller, F., van Gerwen, S., Bange, T., and Musacchio, A. (2017). Basis of catalytic assembly of the mitotic checkpoint complex. *Nature*.

Ji, Z., Gao, H., Jia, L., Li, B., and Yu, H. (2017). A sequential multi-target Mps1 phosphorylation cascade promotes spindle checkpoint signaling. *eLife* 6.

Reviewer #1 (surface plasmon resonance technology expert)

Zhang et al present a biochemical, biophysical and functional analysis of the interactions between human Bub1 and Mad1 and the functional consequences of this interaction. The authors identify the conserved domain 1 (CD1) in Bub1 that is involved in Mad1 direct binding and demonstrate that this interaction is mediated through the phosphorylation of CD1 on Thr461 as well as the hydrophobic residues (Ile471, Met-Phe-Gln: 474-476) from the predicted alpha helical segment. Furthermore the studies demonstrated that the main function of Bub1 is to position Mad1 close to KNL1 MELT repeats. Although the Mad1 interaction with Bub1 was already reported (Genes and Development; 28, 140, 2014), this manuscript is further elucidating this important protein-protein interaction and underlies molecular mechanisms linked to spindle assembly checkpoint signaling at kinetochores in human cells. The data are interesting and novel, but the authors should revise their manuscript and address several specific points given below.

1. There is inconsistency in the peptides and protein constructs that were tested for the binding and functional studies, which needs a clarification and additional

experiments: - In Fig. 3F and H the authors are testing the binding of CD1 peptides using SPR. In the CD1 WT peptide the Met (Fig. 2A) is substituted with Nle amino acid. The authors are not commenting why this substitution was done. Usually Nle is used to probe the role of methionine, but it is important to be determined what the binding affinity of the WT is in order to compare and draw conclusions. This result is missing in the paper.

Our response:

We have used Nle instead of Met to get effective synthesis especially for the phosphorylated peptides. When the protection group is removed from the phosphorylated residues this generates benzyl cations that reacts with the Met sulfur group leading to large amounts of byproducts. We have explained the reason for using Nle in the main text of the revised manuscript and indicated that the Kd with Met might be different than what we are measuring (page 11 lines 3-6 of revised manuscript):

“To achieve a more efficient synthesis of CD1 peptides and in particularly of the phosphorylated peptides we used norleucine (Nle) instead of methionine (Met) but we cannot exclude that Mad1 will have a different affinity for peptides containing methionine.”

2. In addition in Fig. 2B it is showed that four hydrophobic residues I471 and MFQ (474-476) are important for the checkpoint activity. It will be important to be tested if these mutations are also affecting the direct binding of Bub1 to Mad1 which will reveal more insight of the CD1/Mad1 interaction and checkpoint activity. Thus authors should perform binding studies with peptides where these hydrophobic residues are mutated.

Our response:

We thank the reviewer for suggesting this experiment that we have performed accordingly. The I471D mutation and the MFQ/RKK mutations in CD1 prevent Mad1 binding consistent with the fact that they are required for Bub1 function in vivo (New Figure 4A and C).

3. Although using peptides are useful tools for mapping the binding site of the direct interactions and together with mutagenesis studies is allowing characterization of the interactions nature, it is important to be demonstrated that recombinant Bub1 protein has similar binding affinity as the peptide segment. These results will further provide evidence that the CD1 domain is the interacting site.

Our response:

We agree with the reviewer on this. We have tried to use MST to see if we could measure the affinity of Mad1 (597-718) for Bub1 (146-553)/Bub3 but we have not been able to get robust signals this way.

4. For the binding study the authors are using Mad1 construct (597-718) which is 112 residues shorter than the construct used in the follow up functional assays (Mad1 485-718). The authors need to provide explanation why different constructs were used and evidence that they retain similar kinetochore-binding function.

Our response:

We did measure the affinity of Mad1 (485-718)/Mad2 for CD1 peptide phosphorylated on T461 but maybe the reviewer missed this point. The Kd is around 72 μ M compared to 32 μ M for the 597-718 Mad1 (see Fig. 4B,C in revised manuscript and page 11 lines 13-14).

“A recombinant tetrameric Mad1(485-718)/Mad2 complex also bound to the CD1 peptide phosphorylated on Thr461 with a Kd of 72 μ M (Fig. 4B)”.

5. This study is conclusively demonstrating that the direct Mad1-CD1 interaction is stimulated by phosphorylation and furthermore identifies the residues that are phosphorylated and important for the interaction. It was already reported that this interaction is mediated by phosphorylation and authors are discussing this and cite relevant references. Based on the reported studies, in the discussion authors are proposing Cdk1 and Mps1 as potential kinase candidates. Given the quality of the data and to further advance the findings on the phosphorylation role, the authors should extend their studies and use commercially available small-molecule inhibitors of these kinases as chemical tools to test and validate Cdk1 and/or Mps1 involvement in the phosphorylation of CD1 domain. This would provide important insights on the Mad1-CD1 interaction as well as towards elucidating the role of these kinases in regulating the spindle assembly checkpoint signaling. This will increase the novelty of the manuscript and its visibility and interest to researchers in other related disciplines.

Our response:

We have included additional experiments regarding the kinases responsible for phosphorylating CD1. Firstly we have successfully generated a phospho-specific antibody raised against the double phosphorylated CD1 peptide (phosphorylated on S459/T461) and show that this antibody stains prometaphase kinetochores. Thus the phosphorylation(s) on CD1 are present where we anticipate them to be – at kinetochores (New Figure 2 in revised manuscript). Secondly we have performed in vitro kinase assays with recombinant GST Bub1 (425-500) coupled

with mass spectrometry. We phosphorylated with Mps1 or Cdk1-cyclin B1 and can detect phosphorylation of T461 by Mps1 and S459 and T461 by Cdk1-cyclin B1 (New supplementary Figure 4B in revised manuscript). We discuss these new data in light of the recent data from the Yu lab (page 15, lines 20-26 and page 16 lines 1-4).

“An interesting aspect uncovered here is that the CD1 needs to be phosphorylated to function in line with observations from budding and fission yeast^{37,45,58}. We show that this occurs at kinetochores in prometaphase and our in vitro phosphorylation assays implicate Cdk1 and Mps1 as possible kinases. Recent work from the Yu lab suggests that Cdk1 phosphorylates Ser459 to allow subsequent phosphorylation of Thr461 by Mps1⁴⁰. We have not extensively explored this and our data does not exclude this idea but we also see that Mps1 can directly phosphorylate Thr461 in vitro as observed in budding yeast³⁷. Our in vitro phosphorylation assays with Cdk1-Cyclin B1 suggest an alternative possibility namely that Cdk1 phosphorylates both sites and it has been shown that Cdk1 can phosphorylate non-Ser/Thr-Pro sites. Indeed, Thr461 has surrounding residues that are favorable for non-Ser/Thr-Pro sites including a Lysine residue in the +4 position⁵⁹. “

Reviewer #2

The spindle assembly checkpoint prevents gains or losses of chromosomes during their segregation to the daughter cells in mitosis. This manuscript reports an analysis of the significance of an interaction of two crucial components of the checkpoint, Mad1 and Bub1, in human cells. This interaction has been observed before in other organisms (originally by K. Hardwick's laboratory and later by S. Biggins' in *S. cerevisiae*, in papers cited by the authors). The present work focuses on a segment of Bub1, called the Conserved Domain 1 (CD1), and on the important demonstration that this motif interacts with Mad1. Although two very recent papers have reached partly overlapping conclusions (see below), the authors' clearly developed their observations independently. Their observations are important and significantly advance our understanding of the mechanistic basis of checkpoint signaling. The data are generally adequately supporting the conclusions. Overall, this makes for an interesting contribution to the spindle checkpoint field and I support its publication provided that the following points are adequately considered.

Specific points

The introduction, and in particular several claims of priority, will need to be revised in light of recent publication of two papers demonstrating an interaction of

human Mad1-Mad2 and Bub1 (Ji et al. eLife 2017; Faesen et al. Nature 2017. Partial overlap of these two publications with the work discussed here should not be seen, however, as an obstacle towards publication of this article). We also note that another priority claim in the Introduction, that Bub1 is responsible for the precise positioning of Mad1-Mad2, has already been demonstrated (Krenn et al. Current Biology 2014, which the authors cite but in a different context).

Our response:

We have removed priority claims in light of the recent publications and we also cite and discuss the Faesen and Ji papers (see for example page 4 lines 15-16, page 16 line 6).

“In human cells a direct interaction between Mad1 and Bub1 stimulated by Mps1 was recently described^{40,41}”

“We show that the function of Thr461 phosphorylation is to facilitate Mad1 binding, which is fully consistent with recent data from the Yu lab⁴⁰”

We also cite the Krenn et al for proposing that Bub1 can act as a scaffold for SAC complexes on MELT repeats (see page 4 line 16-17):

“and consistently Bub1 has been proposed to scaffold the assembly of SAC complexes on MELT repeats³⁰.”

but want to point out that this study was based on IPs of KNL1 fragments and they did not check if Mad1/Mad2 co-purification was dependent on Bub1 (Figure 3C in Krenn et al).

Page 6, lines 3-4

The authors should give credit to the work of the scholars who first identified the CD1 and ABBA motifs (Meraldi, Yu, Pines, Kops)

Our response:

We credit all these in the introduction where we discuss CD1 and ABBA including the original discovery by Davenport et al. (see page 4 line 5-6 (ref 31-36)) In this specific context we are referring to the role of the Bub1 ABBA motif in recruiting Cdc20 to kinetochores, which is covered in the Pines, and Kops lab papers that we therefore cite here. We thank the reviewer for pointing out that we did not cite the Kops lab paper in the original version of this manuscript.

Page 7, line 5

“A likely explanation for the dominant-negative effect...” Here the argument is

circular. The authors have not demonstrated a dominant-negative effect (which strictly speaking configures an antagonistic relationship to the wild type allele), but rather a reduction in the duration of mitotic arrest in presence of the deltaCD1 construct that is even more pervasive than after depletion of endogenous Bub1. Whether this is due to a dominant-negative effect is unclear at this stage. To prove it, the authors should demonstrate such an effect in the presence of endogenous Bub1.

They may elect to do this experiment, or alternatively refocus the discussion and avoid the confusion that this definition will inevitably arise given that in principle the cells in which these effects are observed ought to have been depleted of endogenous Bub1 (i.e. in cells where no “dominant” effect should be observed). This concern extends to all subsequent instances of “dominant-negative” in the manuscript.

Our response:

We agree with this point by the reviewer and have not used the term dominant negative in the revised manuscript but instead write (page 7 lines 7-10):

“A likely explanation for the inhibitory effect of Bub1 proteins lacking CD1 is that they do not support any SAC signaling and compete with remaining endogenous Bub1 for binding to KNL1 similarly to what has previously been observed with another Bub1 mutant (Bub1 1-311)^{47,48}”

Page 8, lines 12-13

The authors provide no rigorous evidence in this context that the effect they describe is due to lack of phosphorylation of CD1. I agree that this is a reasonable hypothesis, but it is not the only one (e.g. mutations to Ala may be not tolerated). I.e. the authors have not shown a role of phosphorylation, but rather have shown that mutation of two residues that may be phosphorylated provokes the same effect as the deletion of CD1.

Our response:

Strictly speaking the reviewer is right but since the introduction of acidic residues (phosphomimetic mutations) does not inactivate Bub1 this strongly favors that the lack of phosphorylation of Ser459 and Thr461 is likely the reason why the alanine substitutions inactivate Bub1 (we note that in the Ji et al paper they only test the Ala mutation). This result and the fact that we see Mad1 binding being enhanced by Thr461 phosphorylation make us favor this interpretation of the results. It is quite common that the role of phosphorylations is probed by comparing alanine substitutions to that of acidic phosphomimetic substitutions.

Page 9, line 16
Give an angle to that angel

Our response:

Corrected-thanks

Page 10, lines 1-6
These results are nicely consistent with those published by the Yu laboratory in eLife (op. cit.)

Our response:

We point this out in the revised discussion (see page 16 line 6-7).

“We show that the function of Thr461 phosphorylation is to facilitate Mad1 binding, which is fully consistent with recent data from the Yu lab⁴⁰ and the work from budding yeast³⁷.”

Page 10, line 25
When referring to Mad1-Bub1 fusion proteins, the authors should clearly indicate Bub1’s name. The term ‘Mad1 deltaCD1’ is uninformative. Please use Mad1-Bub1(deltaCD1) or equivalent. (Same idea for Mad1-MBR on the next page.)

Our response:

We thank the reviewer for this suggestion that helps clarify the work. We refer now to the fusions as Mad1-Bub1 delCD1 and Mad1-Bub1 (1-280) and have also renamed the Mad2 fusions.

Page 11, lines 13-14
Residues 1-280 of Bub1 contain more than the MELT Binding Region, and therefore the nomenclature is incorrect. The GLEBS motif of Bub1 is sufficient for kinetochore localization and for MELT binding (work of S. Taylor’s and A. Musacchio’s laboratories). In the following lines (15-18) it would be fair to cite previous work in support of the authors’ statement.

Our response:

We thank the reviewer for this comment, which is absolutely correct – please see above where we have renamed the fusion constructs.

Page 13, line 1

'forms' should be in singular form

Our response:

Corrected-thanks

Page 13, line 6

'elusive' is no more.

Our response:

Removed and revised to refer to the recent work from Musacchio and Yu labs (page 15 lines 5-7)

"Here we present experimental data, which unequivocally show a direct Mad1-Bub1 interaction in human cells in agreement with recent studies showing an Mps1 stimulated interaction between the human checkpoint proteins^{40,41}"

Page 13, lines 23-25

The authors should point out here that the phosphorylation site they have described (pT-461) is clearly embedded in a CDK consensus site (SPTV), and that previous work from the Biggins, Musacchio, and Yu laboratories cited above had collectively implicated both Cdk1 and Mps1 in its creation. This is discussed on page 14, but the discussion could be unified in view of the more recent data.

Our response:

We have revised this part of the discussion to incorporate the additional results we have from in vitro kinase assays (Figure 2 and supplementary Fig 4B) and discuss this data in light of the recent work from the Yu lab (see page 15 lines 23-26 and page 16 line 1-4).

"An interesting aspect uncovered here is that the CD1 needs to be phosphorylated to function in line with observations from budding and fission yeast^{37,45,58}. We show that this occurs at kinetochores in prometaphase and our in vitro phosphorylation assays implicate Cdk1 and Mps1 as possible kinases. Recent work from the Yu lab suggests that Cdk1 phosphorylates Ser459 to allow subsequent phosphorylation of Thr461 by Mps1⁴⁰. We have not extensively explored this and our data does not exclude this idea but we also see that Mps1 can directly phosphorylate Thr461 in vitro as observed in budding yeast³⁷. Our in vitro phosphorylation assays with Cdk1-Cyclin B1 suggest an alternative possibility namely that Cdk1 phosphorylates both sites and it has been shown that Cdk1 can phosphorylate non-Ser/Thr-Pro sites. Indeed, Thr461 has

surrounding residues that are favorable for non-Ser/Thr-Pro sites including a Lysine residue in the +4 position⁵⁹. “

Please note that the budding yeast T455 is not part of a Cdk consensus site (PTVT-455) and that Mps1 can directly phosphorylate T455 in vitro (London and Biggins 2014).

Page 13, last line
Please remove claims of novelty

Our response:

Removed

Page 15, line 1
In my opinion ‘independently’ is not the right word, as it implies that these residues can perform a function in the SAC independently of the other motifs and interactions, which is clearly not the case, or at least it is not demonstrated here. I think that the authors mean ‘in addition’.

Our response:

We have changed this sentence to make it more clear (Page 17 lines 4-6)

“Since mutations in the C-terminal globular part of Mad1 are still critical for Mad1 functionality in the Mad1-Bub1 fusion, this argues that these residues are important for an aspect of Mad1 function unrelated to Bub1 binding.”

Figure 1F
Error bars missing

Our response:

This is representative of two independent experiments as indicated in the figure legend.

Figure 3F
If I understand things properly, the ‘control peptide’ is actually a mutant, which is indicated differently in the Table. Can the authors harmonize the description of this peptide?

Our response:

Thanks for pointing this out – we have harmonized the description (Revised

Figure 4C)

In general, the authors could significantly improve usage of space in their figures, making some panels larger and avoiding large white open spaces.

Our response:

We have improved the figures based on this suggestion.

Reviewer #3 (Remarks to the Author):

The spindle checkpoint is a surveillance mechanism that monitors the chromosome attachment with spindle microtubules during mitosis. Until all sister chromatids are properly attached to microtubules, the checkpoint detects unattached kinetochores and generates diffusible “wait signals” to delay the onset of anaphase. A functional spindle checkpoint is critical for eukaryotic cells to prevent chromosome missegregation and aneuploidy. Bub1 is a multifunctional checkpoint kinase that is essential for the spindle checkpoint activation. Previous studies on the non-kinase functions of Bub1 have led to the discoveries of two important motifs in its middle region, the conserved domain 1 (CD1) and the ABBA motif. The ABBA motif of Bub1 contributes to checkpoint activation by recruiting Cdc20 to kinetochores. In contrast, the function of Bub1 CD1 is less understood in human cells. A recent study (Ji et al., 2017) has reconstituted in vitro the phosphorylation-mediated Bub1 CD1 interaction with Mad1 using purified human proteins. Mutations that disrupt Mad1-Bub1 interaction in vitro failed to activate the spindle checkpoint in human cells. But the direct evidence showing that the complex of Mad1-Bub1 indeed exists and is functionally important is still lacking. In this manuscript, the authors have further characterized the function of Bub1 CD1 in binding with Mad1 in human cells. The novel discoveries in this study include: (1) the authors provided the in-vivo evidence of the proximity of Mad1 and Bub1, which requires the RLK motif of Mad1; (2) the authors demonstrated that the function of CD1 is to bring Bub1 and Mad1 in proximity, which can be bypassed by Mad1-Bub1 fusion; (3) the authors revealed that an alpha helix of Bub1, C-terminal to the phosphorylated Ser and Thr residues, was also required for Bub1 function, presumably by contributing to Mad1-Bub1 binding.

Overall, this study has presented several interesting findings that could further our knowledge of the checkpoint function of Bub1, even considering the fact that the binding of Mad1 and Bub1 has been reconstituted in the recently published study (Ji et al., 2017). In particular, the in-vivo evidence showing that Mad1 and Bub1 are in proximity using BirA technique and the finding that Mad1-Bub1 fusion protein can bypass the functional requirement of Bub1 CD1 are very significant.

Thus publication in Nature Communication is recommended. On the other hand, due to the recently published two studies on the in-vitro reconstitution of spindle checkpoint signaling (Faesen et al., 2017; Ji et al., 2017), the manuscript needs a major revision, highlighting the unique discoveries and discussing the two published studies as appropriate. The authors need to address the following major points in order to consolidate some of the major conclusions that are novel in the manuscript. In addition, several minor points need to be addressed as well prior to publication.

Major points

1. In Figure 3C and Figure 5B, the authors showed that Mad1-BirA fusion protein could biotinylate Knl1, Bub1 and BubR1 in an RLK-dependent manner. Considering the fact that the RLK motif is required to establish the direct interaction between Mad1 and Bub1, these lines of evidence therefore imply that Mad1-BirA binds with Bub1 and biotinylates it. To further confirm this inference, it will be informative to test if the biotinylation can be compromised by Mad1-binding deficient mutants of Bub1, such as S459A and T461A. In addition, it is surprising to see Mad2, which forms a constitutive complex with Mad1, is not biotinylated in Figure 3C. Is there any explanation for this? Moreover, were the Mad1-BirA and BirA-Mad1 proteins themselves biotinylated in the assay?

Our response:

We have tried to transfect Bub1 1-553 and Bub1 1-553 del CD1 into our stable cell lines expressing Mad1-BirA to see if we could detect biotinylation of the exogenous Bub1. Unfortunately we cannot detect labeling of Bub1 1-553, which prevents us from determining if the labeling of Bub1 depends on CD1. We have also used Bub1 1-553 -BirA fusions but do not detect labeling of Mad1. Our experience with the BioID approach is that it works well on some proteins, like Mad1, but not others.

We see strong biotinylation of Mad1 (included in the revised manuscript-Figure 3B in new manuscript) but none of Mad2. As biotinylation depends on availability of lysine residues and structural constraints it can be that Mad2 is not labeled for these reasons – we mention this in the revised manuscript (page 10 lines 11-13):

“Despite Mad2 being a stable binding partner of Mad1 we did not detect labeling of Mad2, which can be due to lack of lysine residues that are optimally positioned to be labeled by BirA fused to Mad1.”

2. In Figure 2B, the authors have shown that the alpha helix of Bub1 CD1 is required for Bub1 function. But the mechanism of the alpha helix function was not further pursued. Since this alpha helix has not been characterized previously, the authors might want to provide evidence that the alpha helix contributes to the Mad1-Bub1 binding in order to enhance the novelty of the story.

Our response:

See also our comment to reviewer 1. We have now directly tested the binding of Mad1 to CD1 peptides with mutated I471D and the MFQ/RKK mutations and the mutations prevent Mad1 binding consistent with our original idea (Figure 4A,C in revised manuscript).

3. In Figure 5, the authors pointed out a direct role of Mad1 that is independent of Bub1 binding. This observation is consistent with the two recently published works (Faesen et al., 2017; Ji et al., 2017). Ji et al. suggest that this Bub1-independent function is through a phosphorylation-dependent direct interaction between Mad1 and Cdc20. The authors should test if Mad1 E710/F712 works together with phosphorylated T716 to bind with Cdc20.

Our response:

We have extensively tried to see if we can get purified Cdc20 and Mad1 C-term (597-718) to bind in the presence of Mps1. Despite numerous attempts (n=5) we cannot detect any binding (see example below). Our Mps1 preparation is active based on its ability to phosphorylate a KNL1 fragment and we see clear binding of BubR1 under these conditions. We have analyzed SAC strength in Cdc20 del26-37 and monitored MCC formation and these assays support the amino acids 26-37 of Cdc20 are required for an efficient SAC and MCC formation as suggested by the Yu lab but currently we cannot couple this to Mad1 binding and therefore not test if Mad1 E710/F712 is defective in Cdc20 binding.

Experiment: Recombinant Mad1 597-718 incubated with purified Cdc20 and

samples captured with Cdc20 affinity beads (antibody binds close to the C-box so should not interfere with Mad1 binding to residues 26-37 of Cdc20). BubR1 serves as a positive control. Two different preparations of Mps1 were tested. The reactions were probed for the indicated proteins. The concentration of Cdc20 was 1 μ M and that of Mad1 2 μ M.

Minor points

1. In Figure 4A, it is surprising to see that Mad1-MBR could fully support the checkpoint activity in the absence of endogenous Bub1. How do the authors rationalize the fact that Mad1-MBR bypasses not only the Mad1-binding function of CD1 but also the Cdc20-binding function of ABBA?

Our response:

This also surprised us. Please note that the Bub1 ABBA is not essential for SAC but does make it more efficient. A possibility that we discuss is that there is some redundancy of the C-terminal ABBA motif in BubR1 and the ABBA motif of Bub1 as both contribute to Cdc20 kinetochore localization. A possibility is that in the Mad1-Bub1 fusions the Cdc20 recruited by BubR1 interacts efficiently with Mad1-Bub1 to support SAC signaling (see page 16 lines 18-25 and page 17 lines 1-2). We have tried to test if the C-terminal ABBA motif of BubR1 is critical for SAC signaling when the ABBA motif of Bub1 is missing but these experiments have been technically challenging because we need to deplete both Bub1 and BubR1 and rescue with two constructs and we simply do not get enough cells that could tolerate double knock down and plasmid transfection.

2. It has been showed that the doubly phosphorylated Bub1 CD1 peptide binds with Mad1 stronger than either of the singly phosphorylated Bub1 CD1 peptides (Ji et al., 2017). Thus, the authors should repeat the SPR binding assays in Figure 3F, 3G and 5A with the pS459/pT461 peptide.

Our response:

We have measured the affinity of the double phosphorylated peptide for Mad1 (597-718) and included this in the revised manuscript (Figure 4A, C in revised manuscript). We do not detect a big difference in affinity between the Thr461 phosphorylated peptide and the double phosphorylated peptide (16 μ M versus 32 μ M). Please note that the DP responses measured in the ITC of Ji et al are very low values.

3. Figure 5D. There is no evidence that Bub1 CD1 binds to the head of Mad1 C-terminal domain. According to previous studies, it is more likely that Bub1 CD1 directly binds to the RLK motif of Mad1, which is located at the stem of Mad1 C-terminal domain, not the head. Therefore it is better to draw the Mad1-Bub1

binding in a different configuration.

Our response:

Agree – we have modified the figure accordingly (Figure 6D in revised manuscript)

4. In supplementary Figure 5, the Strep-cy2 could detect all kinetochore proteins that were biotinylated by BirA. It is incorrect for the author to conclude that the BirA fusion proteins localize to kinetochore (Page 34, line 5) based on this data.

Our response:

Agree – changed title of figure legend to reflect this.

5. Page 16, line 18-23. It will be helpful if the authors in this section disclose the amino acid sequences, including the linker regions, of all the fusion proteins used in this study, e.g. Mad1-Bub1, Bub1-Mad2, Mad1-BirA and BirA-Mad1.

Our response:

Have included linker sequences in the main text of the revised manuscript now (page 12 line 12):

“All fusion constructs used here have a flexible six amino acid linker (GSGSGS) between the two proteins”

6. In the BirA-mediated biotinylation assays, it is necessary to show the expression levels of all BirA fused Mad1 proteins comparing the endogenous Mad1.

Our response:

We have included this now in the revised manuscript as supplementary data (Supplementary Figure 5B). We titrated the levels of doxycycline to get close to endogenous levels of expression and use 0,1 ng/ml for induction.

7. In Figure 3H, it was not clear to the reader why the authors used Nle to replace methionine.

Our response:

See also our response to reviewer 1. We have explained this in the revised manuscript.

8. Page 11, line 10. The Mad2 mutations should be Arg133 and Gln134 being mutated to glutamic acid and alanine, respectively.

Our response:

Corrected.

9. Page 30, line 10. “25 mM biotin” seems to be exceeding biotin solubility in water. It might be a typo of “25 uM”.

Our response:

Thanks – corrected.

10. Page 31, line 7. The panel “H” was mislabeled as panel “G”.

Our response:

Thanks - corrected

11. Page 31, line 18. “R133Q/E134A” is a typo. It should be “R133E/Q134A”.

Our response:

Thanks - Corrected

12. Page 32, line 11. “Live cell imaging and quantification were performed as in A)” is referring to the wrong panel.

Our response:

Thanks - corrected

Reviewers' Comments:

Reviewer #1:

Remarks to the Author:

The authors have adequately addressed the significant concerns and strengthened the paper by performing additional experiments and providing new insights on the binding and functional level of the interaction between Mad1 and a segment of Bub1 known as the Conserved Domain 1 (CD1). The manuscript will contribute to the spindle checkpoint field and is suitable for publication in Nature Communications.

Reviewer #2:

Remarks to the Author:

I wish to thank the authors for submitting an improved version of their manuscript. I am happy to support publication of this manuscript in its current form.

Minor point

Line 168

The authors probably refer to Thr461, not Thr471 as written

Reviewer #3:

Remarks to the Author:

The authors have addressed all major and minor concerns raised by reviewers. The revised manuscript has provided several novel findings in the field of spindle checkpoint research. The overall data quality is good, with appropriate citations and discussions of previous studies. Publication in Nature Communication is highly recommended.

REVIEWERS' COMMENTS:

Reviewer #1 (Remarks to the Author):

The authors have adequately addressed the significant concerns and strengthened the paper by performing additional experiments and providing new insights on the binding and functional level of the interaction between Mad1 and a segment of Bub1 known as the Conserved Domain 1 (CD1). The manuscript will contribute to the spindle checkpoint field and is suitable for publication in Nature Communications.

Reviewer #2 (Remarks to the Author):

I wish to thank the authors for submitting an improved version of their manuscript. I am happy to support publication of this manuscript in its current form.

Minor point

Line 168

The authors probably refer to Thr461, not Thr471 as written

Corrected

Reviewer #3 (Remarks to the Author):

The authors have addressed all major and minor concerns raised by reviewers. The revised manuscript has provided several novel findings in the field of spindle checkpoint research. The overall data quality is good, with appropriate citations and discussions of previous studies. Publication in Nature Communication is highly recommended.